# Position: AI Evaluations Should be Grounded on a Theory of Capability

**Nathanael Jo** [1]   **Ashia Wilson** [1]

## Abstract

Evaluations of generative models are now ubiquitous, and their outcomes critically shape public and scientific expectations of AI's capabilities. Yet skepticism about their reliability continues to grow. How can we know that a reported accuracy genuinely reflects a model's underlying performance? Although benchmark results are often presented as direct measurements of capability, in practice they are inferences: treating a score as evidence of capability already presupposes a theory of what it means to be capable at a task.

We argue that AI evaluations should instead be framed as inference tasks grounded on an explicit theory of capability. While this perspective is standard in fields like psychometrics, it remains underdeveloped in AI evaluation, where core assumptions are often left *implicit*. As a proof-of-concept, we empirically show that reported performance can depend strongly on the evaluator's modeling assumptions, underscoring the need for transparent, theory-driven evaluation practices. We conclude by offering an **Evaluation Card** to help researchers document, justify, and scrutinize the modeling decisions underlying AI evaluations.

## 1. Introduction

Evaluations (from hereon, "evals") of generative models using benchmarks have become ubiquitous as a way to probe model capabilities or harms. Companies developing large language models (LLMs) routinely assess their systems' intelligence using standardized knowledge tasks, while research papers proposing new methods often conduct comparative evaluations against state-of-the-art models. Leaderboards hosted on Vellum and Huggingface have also emerged as open-source platforms for directly comparing the capabilities of various LLMs. The rapidly growing inter-

est in evals reflects our collective desire to understand how generative models behave, especially as they are now widely utilized, touted as highly capable, but inherently black box in nature. Yet, there is growing consensus that generative AI evals using benchmarks are brittle and unreliable (Mitchell, 2023; Eriksson et al., 2025). We argue that many of these failures stem from a deeper issue: **AI evals implicitly treat scores as measurements of capability, without specifying what "capability" means in the context of a benchmark.**

To see why this lack of clarity matters, it is useful to return to a more canonical setting in machine learning (ML). Consider a clinical predictive algorithm designed to detect a rare disease. In this case, the notion of *capability* is clear: the algorithm is useful to the extent that it correctly identifies the disease when it is present, and avoids false alarms when it is not, in the population where it will actually be deployed. This naturally motivates appropriate evaluation metrics. Measures such as precision and recall are well suited to this goal, while accuracy is often misleading because it can obscure severe class imbalance. In other words, (good) evaluation in classical predictive settings is tethered to a theory of what the model is meant to do, and metrics are chosen to reflect that theory.

In the era of generative models, this paradigm has shifted substantially because models are often general-purpose. Rather than being evaluated on a single, well-defined task, they are typically assessed using broad benchmarks that aggregate performance across diverse tasks. For example, MMLU-Pro (Wang et al., 2024) is intended to probe general understanding and reasoning, yet individual questions may simultaneously draw on factual recall, linguistic competence, and multiple forms of reasoning. As a result, performance on a benchmark is often implicitly a proxy for some notion of capability that is rarely articulated or agreed upon. When this connection between performance and capability is left implicit, interpreting the resulting conclusions becomes unreliable: it is unclear what a higher score actually means and which abilities have truly improved.

Once a theory of capability has been set, uncertainty quantification becomes unavoidable. Yet most evals do not attempt to quantify the full range of uncertainty inherent in generative models. In predictive ML, uncertainty is relatively straightforward: predictions typically reduce to a

[1]MIT EECS, Cambridge, USA. Correspondence to: Nathanael Jo <nathanjo@mit.edu>.

*Proceedings of the 43rd International Conference on Machine Learning*, Seoul, South Korea. PMLR 306, 2026. Copyright 2026 by the author(s).

low-dimensional probability or score, and the main concern is finite-sample uncertainty arising from limited coverage of the population distribution. Finite-sample uncertainty remains important for generative models (Miller, 2024; Chiang et al., 2024), but these models introduce additional and qualitatively different sources of uncertainty, such as sensitivity to perturbations, context, and inference-time strategies. How should tests be designed to quantify capability under these sources of uncertainty?

**In this position paper, we argue that evals for generative models should return to their roots in inference by starting with a clear definition of capability itself.** We draw inspiration from psychometrics and educational assessments, where statistical models have long been used to infer latent human abilities from observed performance. Following this tradition, our goal is to develop a framework for AI evals: one that makes explicit what notion of capability is being measured and identifies the uncertainties that threaten reliable inference. Doing so allows evaluators to communicate what benchmark results do—and do not—tell us about their true capabilities.

**Contributions.** We make three contributions. First, we outline a suite of possible theories of capability that were derived from *human* cognition. We argue that most AI evals assume an overly simplistic capability theory (Section 2). We then discuss **how these theories may be adapted to account for systematic ways in which AI systems differ from humans** (Section 3).

Second, **we empirically apply these theories of capability as a proof of concept**, showing that they yield systematically different interpretations of the same benchmark results[1] (Section 4). To make this comparison concrete, we study one common source of uncertainty in generative models—sensitivity to input perturbations—but note that our argument is not tied to this choice.

Finally, we emphasize that no theory is strictly better than others; they come with different assumptions and trade-offs on the task structure and data. In Section 5, **we offer an Evaluation Card to help researchers document, justify, and scrutinize the modeling decisions underlying AI evaluations**. We also identify potential avenues for future work in Section 7, and provide an extended related work in Appendix A.

## 2. Current Evals Implicitly Assume a Theory of Capability

In this section, we argue that current evals assume a theory of capability, but fail to make those assumptions explicit.

---

[1]Code can be found in https://github.com/nathanaj99/ai_stat_test.

### 2.1. Most Evals Use a CTT Model

> **Theory 1: Classical Test Theory (CTT)**
>
> Originating in the early 20th century, CTT models an observed test score $\phi$ as the sum of a true score $\theta$ and random error $\epsilon$ (Raykov & Marcoulides, 2011):
>
> $$\phi = \theta + \epsilon. \qquad (1)$$
>
> Assumption 2.1 requires that $\epsilon$ has mean zero and is independent of $\theta$, reflecting the idea that humans make *random, ability-independent* mistakes[a]. Under parallel forms of a test, repeated scores converge to $\theta$ (Lord & Novick, 2008).
>
> **Assumption 2.1.** Under the CTT model in (1), $\mathbb{E}[\epsilon] = 0$ and $\mathrm{Cov}(\theta, \epsilon) = 0$.
>
> ---
> [a]CTT additionally assumes independence of errors across questions, which we ignore here since AI typically exhibit this property.

In practice, CTT reduces to averaging correctness across items, implicitly assuming that all questions are equally informative about capability. This mirrors nearly all contemporary AI evals, which report simple aggregate statistics on performance. Formally, letting

$$\phi_i = \theta_i + \epsilon_i, \qquad (2)$$

CTT treats each observed response as an unbiased but noisy measurement of a per-item score $\theta_i$, yielding overall capability $\theta = \mathbb{E}_i[\theta_i]$. This assumption underlies virtually all widely used static benchmarks across NLP, vision, and reasoning tasks: a model's "capability" is its average performance under the question distribution. **We note that CTT need not only capture accuracy** or correctness; for example, HELM (Liang et al., 2023) reports metrics such as fairness and calibration, which translate to redefining $\phi_i$:

$$\phi_i = \mathrm{disp}_i(\hat{y}, y, a); \quad \phi_i = |\Pr(y = 1|\hat{p} \in B_i) - \hat{p}|,$$

where $\hat{y}, y$ are predicted and true outcomes, $a$ is protected group, disp is some disparity metric, $\hat{p}$ is the predicted probability, and $B_i$ denotes a bin of predicted confidence values.

### 2.2. Recent IRT-Based Methods Implicitly Adopt a Different Theory of Capability

A growing line of work proposes IRT-inspired methods for AI evaluation, often to reduce sample complexity by leveraging question-level information (e.g., difficulty). These approaches replace average performance (CTT) with a *latent trait* model in which an ability parameter $\theta$ generates response probabilities.

---

Theory 2: Item Response Theory (IRT)

Formally, let $\theta \in \mathbb{R}^K$ be a $K$-dimensional ability vector that governs the probability of answering correctly. The most common model specifies

$$f_i(\theta) = \Pr(\phi_i = 1 \mid \theta) = \sigma(a_i^\top \theta - b_i), \quad (3)$$

where $a_i \in \mathbb{R}^K$ are item loadings and $b_i \in \mathbb{R}$ is item difficulty. Intuitively, Eq (3) says that the probability of getting a correct answer for question $i$ depends on one's ability $\theta$, how ability interacts with the question $a_i$, subtracted by how difficult that question is $b_i$. More capable models might still have trouble with more difficult questions. We model observed responses as Bernoulli draws,

$$\phi_i = f_i(\theta) + \epsilon_i, \quad (4)$$

where $\epsilon_i$ represents logistic or probit noise. By modeling how items discriminate along dimensions in $\theta$, IRT provides sample-efficient estimates, and underlies much of today's adaptive standardized tests (College Board, 2025). IRT also satisfies Assumption 2.1; see Proposition B.1.

---

IRT originated from psychometric testing of humans, but it is now the bedrock for contemporary AI evaluations. Examples include an adaptive testing design (Zhuang et al., 2023), constructing smaller, more informative benchmarks (Maia Polo et al., 2024), and others (Martínez-Plumed et al., 2019; Rodriguez et al., 2022; Schilling-Wilhelmi et al., 2025; Xu et al., 2025; Zhou et al., 2025; Castleman et al., 2025; Jo et al., 2026). While powerful, these works instantiate a *different* theory of capability: ability $\theta$ is a latent parameter defined by a specific generative model, not the empirical accuracy. Under IRT, two models with the same accuracy can receive different ability estimates if their errors fall on items with different discrimination parameters. Without making the underlying theory explicit, such differences in what "capability" can easily lead to misinterpretation, as we show empirically in Section 4.3.

## 3. Rethinking Capability for AI Systems

The CTT and IRT models are certainly reasonable starting points for the emergent field of AI evaluations. In this section, we outline other theories of capability that may better reflect AI's emergent abilities. However, since all these models were developed to describe *human* behavior, we argue that these theories must be adapted to account for the unique ways AI introduces uncertainty.

### 3.1. Alternative Theories of Capability

We summarize the structural assumptions across various theories of capability in Figure 1.

---

Theory 3: Cognitive Diagnostic Models (CDM)

CDMs represent capability as a vector of (typically) discrete skill masteries (Leighton & Gierl, 2007). Let $S \in \{0,1\}^K$ encode whether a test-taker has mastered each of $K$ underlying skills, and let $Q \in \{0,1\}^{m \times K}$ map items to the skills they require. In Figure 1(c), we have $K = 3$ skills and $m = 3$ questions, and answering question $X_1$ correctly requires skills $S_1$ and $S_2$. A simple example is that $X_1$ is the question $3 + 4 - 1$, which requires two skills: $S_1$ (addition) and $S_2$ (subtraction). Concretely, the probability of correctness is given by

$$\Pr(Y_{ij} = 1 \mid S_j) = \begin{cases} 1 - s_i, & \eta_{ij} = 1, \\ g_i, & \eta_{ij} = 0. \end{cases}$$

Here $s_i$ is the *slip probability*: even when the respondent has all required skills ($\eta_{ij} = 1$), they may answer incorrectly due to inattention or randomness. Conversely, $g_i$ is the *guess probability*: even without the full skill set, the respondent may answer correctly by chance or via a shortcut.

---

A central modeling choice in CDMs is how to aggregate the mastery vector to determine $\eta$. There are generally two choices:

*(i) DINA (AND gate).* $\eta_{ij} := \prod_{k=1}^{K} S_{jk}^{Q_{ik}}$

*(ii) DINO (OR gate).* $\eta_{ij} := 1 - \prod_{k=1}^{K} (1 - S_{jk})^{Q_{ik}}$

Intuitively, DINA assumes that an item can be solved only if *all* required skills are mastered. DINO, on the other hand, assumes that possessing *any one* of the skills may be sufficient to answer correctly (e.g., there exists multiple solution strategies).

---

Theory 4: Bayesian Network Skill Models (BNSM)

In intelligent tutoring systems, capability is often modeled as a structured latent state evolving over a graph of prerequisite relations. Skills form nodes of a Bayesian network, and each skill variable $S_k$ has a posterior $\Pr(S_k = 1 \mid \text{data})$ updated by item responses (Culbertson, 2016). Items provide noisy evidence via conditional probability distributions (CPDs), and inference propagates beliefs across the graph. This yields a capability representation that is *structured*, accommodating both conceptual dependencies and temporal updates.

---

| **(i) Classical Test Theory (CTT)** | **(ii) Item Response Theory (IRT)** | **(iii) Cognitive Diagnostic Model (CDM)** | **(iv) Bayesian Network Skill Model (BNSM)** |
|---|---|---|---|
| 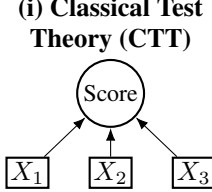 | 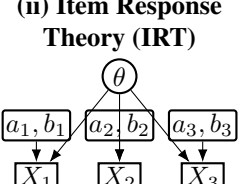 | 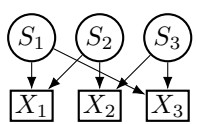 | 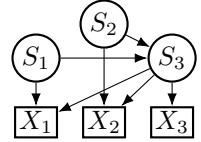 |
| Observed responses to questions $X_i$ aggregate into an overall test score. | Latent ability $\theta$ generates responses as a function of item parameters $(a_i, b_i)$ (e.g., discrimination and difficulty). | Discrete skills $\{S_k\}$ (e.g., spelling, reasoning) generate responses, depending on which questions require which skills. | Skills form a directed hierarchy; mastery of some skills depends on others, and questions depend on certain skills. |

*Figure 1.* Structural assumptions across various theories of capability.

---

**Theory 5: Response time models (RT)**

RT models not only assess whether a test-taker answers an item correctly, but also *how long* they take to respond. The central idea is that response times carry information about latent cognitive processes such as fluency, effort, or deliberation. For item $i$ and respondent $j$, let $T_{ij} > 0$ denote the observed response time. A widely used model (Van der Linden, 2007) posits

$$\log T_{ij} \;=\; \tau_i \;-\; \phi_i \zeta_j \;+\; \varepsilon_{ij}, \quad \varepsilon_{ij} \sim \mathcal{N}(0, \sigma_i^2). \tag{5}$$

Here $\zeta_j$ is a *speed* parameter for respondent $j$, $\tau_i$ captures *time intensity* (how long the item typically takes to solve), and $\phi_i$ is a *time discrimination* parameter controlling how sensitive response time is to differences in speed. $\varepsilon_{ij}$ is a noise term. Response time models are often coupled with a standard item response model (e.g., CTT or IRT), with the latent speed $\zeta_j$ allowed to correlate with ability.

---

We emphasize that this list is non-exhaustive. For example, there are nonparametric item response models such as Mokken scaling that drop parametric assumptions (Mokken, 1971; Sijtsma & Molenaar, 2012) and dynamic learning models such as Bayesian Knowledge Tracing and performance factor analysis that treat capability as evolving over time (Corbett & Anderson, 1994; Pavlik et al., 2009). In psychophysics and decision science, ability is operationalized via signal detection sensitivity or drift-diffusion parameters (Green & Swets, 1966; Ratcliff & McKoon, 2008).

### 3.2. AI Introduces Systematic Variation

In general, theories of capability share a common structure: latent ability generates some performance, and deviations from that structure are treated as random, independent across items, and independent of ability itself. In CTT this is explicit through Assumption 2.1; in IRT, CDM, and BNSM,

it appears through the assumption that conditional on the latent trait, item responses are independent and error terms are unbiased. These assumptions are somewhat reasonable for human cognition, but they are **systematically violated by current AI systems.**

A large body of evidence shows that AI models often fail in ways that humans do not: they exhibit "Potemkin" or superficial understanding (Mancoridis et al., 2025), brittle semantic generalization (Mizrahi et al., 2024; Sclar et al., 2023; Zheng et al., 2023), and sensitivity to superficial rephrasings, distractors, and formatting (Zhuo et al., 2024; Errica et al., 2024; Du et al., 2022). Moreover, model behavior depends strongly on hyperparameters (e.g., temperature, top-$p$), inference-time strategies, and surrounding conversational context. These behaviors contradict the foundational assumption shared across human-centric capability theories: that errors represent noise rather than systematic, context-driven shifts in performance.

To illustrate, consider a more realistic model of AI capability *within the CTT paradigm*:

$$\phi_i = \theta_i + s(x_i) + r(h) + g(c) + \cdots + \epsilon_i, \tag{6}$$

where $x_i$ captures input features of item $i$ (e.g., phrasing or structure), $h$ denotes hyperparameters (e.g., sampling temperature), and $c$ encodes contextual or environmental variables (e.g., system prompts). The functions $s$, $r$, and $g$ represent *systematic*, non-random performance shifts. When $x_i$, $h$, or $c$ vary across evaluation settings, these structured biases confound estimation of the underlying item-level capability $\theta_i$. The ellipsis indicates additional confounding factors. **We emphasize that all capability theories can be similarly adapted**, see Table 1 for an example.

### 3.3. Agentic and Reasoning Capabilities

Reconsidering capability becomes more complex for ever-evolving AI systems. Consider, for instance, the response time (RT) model from Section 3.1. In human testing, re-

sponse time is informative because it reflects internal cognitive processes such as deliberation, effort, or speed-accuracy tradeoffs, and is therefore meaningfully coupled with latent ability. For most contemporary AI systems, however, this assumption fails: latency in responses is often dominated by external factors such as hardware constraints, batching, or network delays, which are largely orthogonal to the model's ability. As a result, RT does not play the same diagnostic role for AI as it does for human test takers.

That said, recent agentic or reasoning-enabled systems can explicitly choose when and how much to "think": for example, by allocating additional computation or producing intermediate reasoning steps for harder tasks. In such settings, a notion of response time—or more precisely, *deliberation length or compute usage*—may once again become informative. Crucially, however, this form of "time" differs from its human analogue. Let $T_{ij} > 0$ denote a measure of *computational expenditure* when model $j$ answers item $i$, such as wall-clock latency, number of reasoning tokens, or FLOPs. We posit a log-normal model analogous to Eq. (5):

$$\log T_{ij} = \tau_i + \phi_i \zeta_j + \beta^\top Z_{ij} + \varepsilon_{ij}, \quad \varepsilon_{ij} \sim \mathcal{N}(0, \sigma_i^2).$$

Here $\zeta_j$ is a model-specific *deliberation propensity* capturing how aggressively the system allocates compute across tasks, $\tau_i$ captures the extent to which item $i$ elicits additional effort, and $\phi_i$ governs sensitivity to variation in deliberation propensity. The additional term $\beta^\top Z_{ij}$ absorbs structured systems-level contributions to latency (e.g., hardware, batch size, server load, or decoding settings).

In this model, $T_{ij}$ does not reflect cognitive processes, but is a product of strategic or architectural choices shaped by prompting, system policies, and resource constraints. This may be particularly useful for evaluators to analyze agentic AI systems' speed-accuracy tradeoffs, or their tendency to expend more computation than necessary.

## 4. Proof-of-Concept: Input Sensitivity

As before, we emphasize that our goal is not to identify a *correct* theory of capability, but to show how different theories impose different assumptions and yield incomparable results. In this section, we will operationalize this idea by addressing one issue that confounds evaluations: sensitivity to input perturbations. We proceed in two parts: *(1)* start by specifying a theory of capability (here, we outline four); then *(2)* from those theories, derive inference strategies to estimate the latent ability construct.

### 4.1. Part 1: Theories of AI Capability

Table 1 presents several examples of theories of capability. Each theory requires Assumption 4.1, which states that capability is recovered only *in expectation* over the distri-

bution of "natural perturbations". Without this assumption, latent traits become unidentifiable: the perturbation function $s(\cdot)$ can absorb item properties, trait parameters, or both. **Explicitly stating the theory of capability makes these crucial assumptions transparent.**

| Model family | Functional form |
|---|---|
| **CTT** | $\phi_i = \theta_i + s(x_i) + \epsilon_i$ |
| **IRT (1-dim)** | $\phi_i = \sigma(\theta - b_i) + s(x_i) + \epsilon_i$ |
| **CDM** | $\phi_i = f_{\text{CDM}}(\alpha, Q_i) + s(x_i) + \epsilon_i$ |
| **BNSM** | $\phi_i = \Pr(\phi_i = 1 \mid S, \text{graph}) + s(x_i) + \epsilon_i$ |

*Table 1.* Functional forms of four capability theories, augmented with a perturbation term $s(x_i)$ capturing systematic shifts due to variations in input phrasing or structure.

**Assumption 4.1** (Mean-zero perturbations). Let $\mathcal{P}_i$ denote the distribution of natural perturbations of question $i$. Then

$$\mathbb{E}_{x_i \sim \mathcal{P}_i}[s(x_i)] = 0.$$

**Benchmark Curation Violates an Independence Assumption.** We focus on the CTT model for clarity, though a similar argument holds trivially for the other models in Table 1. Let $\mathcal{D} = \{x_i\}_{i=1}^n$ denote a benchmark. Conceptually, generating an item $x_i$ involves two stages:

*Stage 1 (Question sampling):* Draw a question or concept $i$ from a latent distribution over the task space $\mathbb{P}$.

*Stage 2 (Phrasing sampling):* Draw a natural phrasing $x_i$ for that question from the high-dimensional, unknown phrasing distribution $\mathcal{P}_i$.

Benchmark curators effectively control Stage 1 through their choice of questions, which can be viewed as independently sampled from an implicitly defined $\mathbb{P}$. However, curators do *not* observe or sample from the true $\mathcal{P}_i$. In practice, each question receives only a single, hand-designed phrasing $x_i$, and these phrasings are produced by the same individuals or pipeline, introducing stylistic and structural dependencies across items. Thus, benchmarks almost always produce *dependently sampled* draws from $\mathcal{P}_i$, violating Assumption 4.1. This makes it impossible to identify $\theta_i$ under Table 1 (top row), even though identification is trivial under the classical CTT model (2). Empirically, this manifests as different or conflicting inferences on accuracy, see Appendix C.

**Perturbations for Pseudo-Independence.** A natural response to the dependence problem is to approximate $\mathcal{P}_i$ by generating multiple phrasings for each question. Prior work already follows this intuition: perturbing instructions (Mizrahi et al., 2024), question wording (Sclar et al., 2023), or answer ordering (Zheng et al., 2023) all implicitly aim to sample from a richer portion of $\mathcal{P}_i$. Our framework clarifies

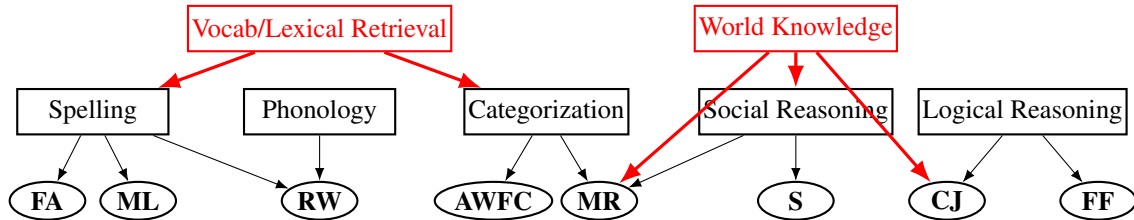

*Figure 2.* Skill structure for CDM (DINA) and BNSM. Black arrows show the skill–task mapping. Red nodes/arrows show additional latent structure used in BNSM (World Knowledge $W$ and Retrieval $R$).

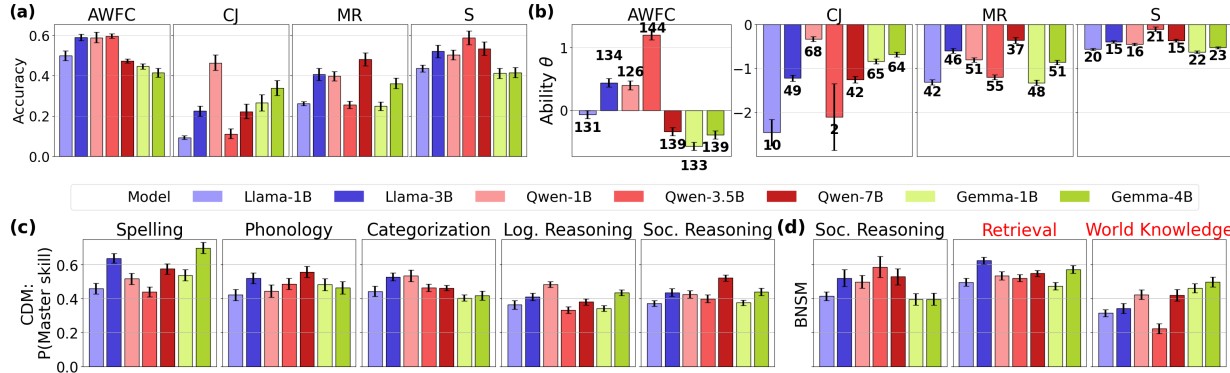

*Figure 3.* (a) Estimates of accuracy under CTT (Alg. 1), (b) Estimates of ability under IRT (Alg. 2), (c) and (d) Estimates of mastery of skills under CDM (Alg. 3) and BNSM (Alg. 4), respectively, as specified by the structure in Fig. 2. Red text indicates skills defined only in BNSM. These theories and methods explicitly account for model sensitivity to perturbations, see Table 1. We test seven open-source LLMs on eight benchmark tasks across LMEntry and BBH. Numbers in bold indicate number of questions asked in the adaptive test. Only three BNSM skills are shown here for brevity since the other skills have similar inferences to CDM. Also for brevity, only four tasks are shown for CTT and IRT. See Appendix E for full results.

that these methods are attempts to recover identifiability by increasing coverage of the phrasing space.

Let $\tilde{\mathcal{D}} = \{\{x_{ij}\}_{j=1}^{m_i}\}_{i=1}^{n}$ be a perturbed benchmark, where each $x_{ij}$ is produced by a perturbation mechanism intended to approximate draws from $\mathcal{P}_i$. The CTT model then becomes $\phi_{ij} = \theta_i + s(x_{ij}) + \epsilon_{ij}$. Although perturbation generators can never match the true (and fundamentally unknowable) $\mathcal{P}_i$, perturbations may improve identifiability, as in Proposition B.3. Regardless, the perturbation mechanism remains a modeling decision.

### 4.2. Part 2: Inference Strategies

Once a theory of capability is fixed, inference strategies often follow from standard statistical results. In this section, we demonstrate how different theories of capability naturally induce different estimands and inference pipelines.

**CTT.** To estimate $\theta$ under prompt perturbations, we use a *clustered bootstrap*, treating items as clusters and perturbations as observations within clusters (Ren et al., 2010; Field & Welsh, 2007). This yields uncertainty over average accuracy without explicitly modeling variance. See Algorithm 1 in Appendix D.

**IRT.** Given item parameters $(a_i, b_i)$, estimating ability $\theta$ re-

duces to classical IRT inference via Fisher scoring / Newton-Raphson updates (Raykov & Marcoulides, 2011). Prompt perturbations are handled as clustered observations per item, and adaptive item selection can be used to reduce sample complexity. See Algorithm 2 in Appendix D.

**CDM.** We relax the binary skill assumption and allow $\alpha \in \mathbb{R}^K$. Inference then reduces to penalized maximum a posteriori (MAP) estimation under a logistic likelihood and a Gaussian prior. Uncertainty is obtained via item-level bootstrapping. See Algorithm 3 in Appendix D.

**BNSM.** We model skills as correlated continuous latent variables in a Bayesian network. Aggregated perturbations provide item-level evidence, and posterior inference yields calibrated skill estimates with bootstrap uncertainty. See Algorithm 4 in Appendix D.

### 4.3. Empirical Study

**Setup.** We evaluate seven open-source instruction-tuned LLMs (Llama-3.2, Qwen-2.5, and Gemma) on two benchmarks, Big-Bench Hard (BBH) (Suzun et al., 2023) and LMEntry (Efrat et al., 2023), both of which have perturbed versions from (Mizrahi et al., 2024). Each dataset contains sub-tasks testing different concepts, and we use four from

each category. For LMEntry, we use `any word from category` (**AWFC**), `first alphabetically` (**FA**), `more letters` (**ML**), and `rhyming word` (**RW**). For BBH, we use `causal judgment` (**CJ**), `movie recommendation` (**MR**), `formal fallacies` (**FF**), and `snarks` (**S**). See Appendix C.1 for details on the tasks, perturbations, and evaluation procedure.

**Skill structure.** For CDM and BNSM, we specify a skill mapping in Figure 2, assuming a DINA model: an item is answered correctly only if *all* required skills are mastered. This is a simple choice of many, but serves as a demonstration of the flexibility of CDM and BDSM.

**Results.** Figure 3 show estimates of capability from the four methods introduced in Section 4.2 over seven LLMs. We show the full suite of results in Appendix E. Generally, the model rankings are consistent between CTT and IRT. However, IRT yields more separation between models when CTT cannot, and using fewer samples (in **bold**). For example, `Qwen-3.5B` on **AWFC** benchmark has the highest inferred performance using both methods, but IRT infers much higher ability for that model compared to the rest because the model performs well on harder questions. We also observe that high accuracy (CTT) does not necessarily translate to high ability (IRT). `Qwen-3.5B` achieves the same accuracy on **AWFC** and **S**, but has a lower ability score on **S**. This is because **AWFC** comprises harder questions, and performing well on harder questions indicates higher ability.

Meanwhile, CDM and BNSM can both report *skill-level* capability by pooling evidence across tasks according to the assumed skill structure (Figure 2). While most of the inferences are similar, BNSM infers markedly higher Social Reasoning scores ($Soc$) compared to CDM. This is because, in CDM, the only way to explain poor performance on **MR** is to reduce the mastery of the $Soc$ skill. In contrast, BNSM introduces an additional latent skill World Knowledge ($W$), which loads directly on **MR, CJ**, and also influences $Soc$. As a result, when **MR, CJ** are systematically low performing while **S** provides comparatively stronger evidence for $Soc$, the posterior in BNSM can attribute a larger fraction of the **MR/CJ** errors to low $W$ rather than to low $Soc$.

## 5. Guidelines for Evaluators

We have shown that evaluators have significant degrees of freedom to choose a theory of capability for AI evaluations. In the face of such discretion, we propose that evaluators should carefully justify and report the following decisions in an **Evaluation Card.**

In Table 2, we illustrate how different theories of capability we outlined in this paper align with the modeling assumptions and decisions evaluators must make. We note, however, that these theories are by no means an exhaustive list.

---

**Evaluation Card**

### 1. Meaning of Capability.
*What should a higher score to mean?*
Different theories define different capability constructs, which are not directly comparable. For example, while CTT simply reports average performance, CDM/BNSM yields an estimate of mastery over a set of researcher-specified skills.

---

### 2. Task or Domain Structure.
*Do tasks admit a meaningful latent structure, and if so, what kind?*
For example, a benchmark might contain questions that implicitly test different sets of skills, such as a combination of spelling, phonology, and causal reasoning. Simply reporting accuracy (under CTT) or ability (under IRT) without imposing structure on the types of skills each question tests might obscure a natural interpretation of benchmark performance.

---

### 3. Sources of Systematic Variation.
*Which factors are averaged over or held fixed?*
Evaluators must decide which sources of variation are explicitly modeled, and which sources are noise or conditioned upon. For example, in Section 4, we explicitly accounted for finite-sample uncertainty and sensitivity to perturbations. However, we conditioned on the same inference-time strategy (i.e., constant hyperparameters and no best-of-N sampling or reasoning) and contextual environment (i.e., no influence of long prior chats).

---

### 4. Robustness Checks.
Where feasible, conduct evaluations under other plausible theories of capability to assess whether the resulting inferences remain stable across different assumptions about model capabilities. When results differ, identify the discrepancies and propose possible explanations for them.

---

## 6. Alternative Views

*Alternative View 1:* **Benchmarks as engineering heuristics rather than inference tasks.** One alternative view is that benchmarks should primarily support rapid iteration, rather than as thorough, statistical inference tasks. This view aligns with prior works advocating that benchmarks should function as operational tools embedded in complex ML pipelines, where simplicity and standardization are critical for joint progress (Snoek et al., 2018; Dodge et al., 2019). Accuracy-based aggregation is attractive precisely because it is easy to compute and communicate. Thus, ambiguity

| Aspect | Implication on choice of theory |
|---|---|
| 1. Meaning of Capability. What should a higher score mean? *Scores from different theories represent distinct constructs and are not directly comparable.* | **CTT:** average performance on a fixed item set. **IRT:** latent continuous ability relative to item parameters. **CDM/BNSM:** mastery of interpretable skills. **RT:** effort (e.g., reasoning) is also important on top of ability |
| 2. Task or Domain Structure. Do tasks admit a meaningful latent structure? *Imposing unjustified structure risks misspecification.* | **No structure:** treat items as exchangeable (CTT). **Items vary in difficulty and discrimination:** (IRT). **Items test varying skills:** (CDM/BNSM).    *All skills required to solve question*: DINA    *At least one skill required to solve question*: DINO |
| 3. Sources of Systematic Variation. Which sources of variation are noise vs. structure? *Evaluators must decide which factors are absorbed, modeled, or conditioned upon.* | **Humans:** phrasing, context, and other factors can be treated as mean-zero noise. **AI systems:** exhibit systematic variation from prompts, context, hyperparameters, inference-time methods, etc. |
| Data considerations may restrict choices. What is realistically identifiable with available data? *Richer theories trade data complexity for stronger assumptions.* | **CTT:** no additional data required **IRT:** requires calibrated item parameters, but enables adaptive testing to reduce sample complexity. **CDM/BNSM:** requires strong priors on item-to-skill mapping or dependency graph. |

*Table 2.* Practical considerations for choosing a theory of capability. Each aspect corresponds to a modeling commitment that determines how evaluation scores should be interpreted and compared.

about what is being measured is seen as an acceptable trade-off so long as the benchmark correlates with downstream performance or developer intuition.

**Our response.** This perspective is directly in tension with our emphasis on explicit theories of capability. Our response is not that heuristic benchmarks are illegitimate, but that their use implicitly commits evaluators to assumptions about what variation is meaningful versus noise. Making those assumptions explicit need not preclude fast iteration, but can improve transparency and reduce misinterpretation when benchmark results are used beyond their original context.

*Alternative View 2:* **Prioritizing downstream validity for benchmarks.** A second view prioritizes *downstream validity* over the interpretability of evaluation scores. Under this perspective, the primary goal of benchmarking is to predict real-world outcomes—such as deployment failures, safety incidents, or user satisfaction—rather than to recover latent skills with interpretable meaning. This position reflects longstanding concerns that benchmark scores often fail to transfer cleanly to real-world use, motivating calls for evaluations that better reflect deployment conditions and external validity (Raji et al., 2021; Gebru et al., 2021).

**Our response.** We agree that internal interpretability should *not replace* downstream validity, but the two should serve complementary roles. When benchmark scores are used to make scientific claims about progress, unclear inferences can obscure failure modes and inflate confidence. Explicit capability models can clarify what is—and is not—being predicted, even when downstream performance remains the ultimate arbiter of success.

# 7. Future Work and Research Directions

We conclude by outlining several research directions in the emerging science of AI evaluation.

**A taxonomy of confounders.** In Section 3, we posited several sources of variation that confound evaluations, such as hyperparameters, inference-time strategies, and prompt formatting. However, this is by no means a comprehensive list. Developing a systematic taxonomy of these confounders would help researchers more clearly articulate and reason about the sources of uncertainty their evaluations must address. More broadly, this suggests viewing benchmarks not as fixed datasets, but as experimental protocols whose degrees of freedom must be explicitly modeled.

**Norms for evaluation claims and reuse.** Our work points to a need for clearer norms around how evaluation results are reported and reused. Benchmark scores are frequently repurposed beyond their original context—to make claims about generality, safety, or progress—without revisiting the assumptions under which they were produced. While these norms have been partially established in the predictive setting (Gebru et al., 2021), more work is needed to update these norms in light of of general-purpose models and modern benchmarking practices – perhaps using the lessons we established in Section 5 as a starting point.

**Evolving theories of capability for generative models.** As we have argued throughout, choosing a theory of capability implicitly encodes assumptions about how AI systems behave. These assumptions were historically motivated by human cognition—not generative models—and may be misspecified or incomplete. As our understanding of model

behavior improves, so too must the underlying theories. For example, defining constructs such as item difficulty or latent ability raises philosophical questions: what does it mean for a problem to be intrinsically difficult for an AI system, and should difficulty be benchmark-relative or universal?

Recent work suggests the possibility of a uni-dimensional intelligence factor for LLMs (Ilić & Gignac, 2024), prompting new questions about how such a quantity should be tested and robustly measured. We view our framework as contributing to a broader effort to build the science of benchmarks (Hardt, 2025) and the measurement theory of artificial general intelligence (Mitchell, 2024).

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

## A. Other Related Work

**Perspectives on AI benchmarking.** A broad literature highlights conceptual and methodological gaps in how AI evaluations are designed and interpreted. Psychometrics provides a mature foundation for formalizing constructs such as *validity* – that evaluations properly measure a construct of capability – and *reliability* – that evaluations yield measures that are replicable and consistent (Lord, 1980; Raykov & Marcoulides, 2011). Several recent works argue that AI benchmarking would benefit from similar principles (Wang et al., 2023; Raji et al., 2021), while others critique how benchmarks are built, saturated, and deployed (Kiela et al., 2021; Bowman, 2023; Dehghani et al., 2021). Our contribution extends this line of work by making capability assumptions explicit, modeling the sources of uncertainty that undermine validity, and treating benchmark evaluation as a principled inference task.

**Skills and artificial general intelligence.** There is increasing interest in quantifying artificial general intelligence, which could mean either high performance across diverse tasks (Hernández-Orallo et al., 2021) or the ability to accumulate and recombine skills (Chollet, 2019). Many probes on this question often focus on large composite benchmarks such as MMLU (Hendrycks et al., 2021) and AGIEval (Zhong et al., 2023), without consideration on how specific questions or tasks map into some latent notion of intelligence. Our framework clarifies how different capability models encode a particular skill structure, offering a principled link between benchmark questions/tasks and latent skill hierarchies relevant to discussions of artificial general intelligence.

**Robustness.** Across modalities, modern AI systems exhibit marked sensitivity to even *non-adversarial* perturbations. In vision, small semantic-preserving changes can shift predictions due to reliance on non-robust features (Ilyas et al., 2019); in language, LLMs vary unexpectedly under prompt rephrasings, formatting changes, or answer-order variations (Zheng et al., 2023; Sclar et al., 2023; Mizrahi et al., 2024; Zhuo et al., 2024). While related, we make a broader argument that there is mismatch between how AI models behave in testing environments and the behavioral assumptions inherited from human-centered evaluation theory – many of which involves robustness issues.

## B. Propositions and Proofs

**Proposition B.1.** *Under the IRT model* (3)–(4)*, the residuals* $\epsilon_i = \phi_i - f_i(\theta)$ *satisfy*

$$\mathbb{E}[\epsilon_i] = 0, \qquad \text{Cov}(\theta, \epsilon_i) = 0.$$

*Hence IRT implicitly meets Assumption 2.1.*

*Proof.* The key to this proof is that and assuming *local independence* (i.e. the responses $\{\phi_i\}$ are conditionally independent given $\theta$),

By definition of the model,

$$\phi_i = f_i(\theta) + \epsilon_i,$$

so equivalently

$$\epsilon_i = \phi_i - f_i(\theta).$$

1. **Zero mean.** Since $\phi_i \mid \theta \sim \text{Bernoulli}(f_i(\theta))$,

$$\mathbb{E}[\epsilon_i \mid \theta] = \mathbb{E}[\phi_i \mid \theta] - f_i(\theta) = f_i(\theta) - f_i(\theta) = 0.$$

Taking the outer expectation gives

$$\mathbb{E}[\epsilon_i] = \mathbb{E}[\mathbb{E}[\epsilon_i \mid \theta]] = 0.$$

2. **Zero covariance.** We compute

$$\text{Cov}(\theta, \epsilon_i) = \mathbb{E}[(\theta - \mathbb{E}[\theta]) \epsilon_i] = \mathbb{E}[\mathbb{E}[(\theta - \mathbb{E}[\theta]) \epsilon_i \mid \theta]].$$

Inside the inner expectation, $\theta$ is fixed, so

$$\mathbb{E}[(\theta - \mathbb{E}[\theta]) \epsilon_i \mid \theta] = (\theta - \mathbb{E}[\theta]) \mathbb{E}[\epsilon_i \mid \theta] = (\theta - \mathbb{E}[\theta]) \cdot 0 = 0.$$

Hence $\text{Cov}(\theta, \epsilon_i) = 0$.

$\square$

**Proposition B.2.** *If $x_{ij}$ is independently sampled from $\mathcal{P}_i$, then define an estimator $\hat{\theta}_i = \frac{1}{m_i} \sum_{j=1}^{m_i} \phi_{ij}$. By Assumption 4.1, we have $\hat{\theta}_i \xrightarrow{as} \theta_i$ as $m_i \to \infty$*

*Proof.* Taking expectations,

$$\mathbb{E}[\phi_{ij}] = \theta_i + \mathbb{E}[s(x_{ij})] + \mathbb{E}[\epsilon_{ij}] \stackrel{\text{Asn. 4.1}}{=} \theta_i + 0 + 0 = \theta_i.$$

Since the $\phi_{ij}$ are (pseudo-)independent draws with finite mean, the Strong Law of Large Numbers gives

$$\hat{\theta}_i = \frac{1}{m_i} \sum_{j=1}^{m_i} \phi_{ij} \xrightarrow{as} \mathbb{E}[\phi_{ij}] = \theta_i,$$

as $m_i \to \infty$. $\qquad\square$

**Proposition B.3.** *Let a benchmark contain phrasings drawn from an unknown distribution $\mathcal{P}_i^{(0)}$. Let $\delta_i^{(0)} := \mathbb{E}_{\mathcal{P}_i^{(0)}}[s(x)]$ denote the induced bias in the recovered latent trait. Suppose a perturbation mechanism generates $m_i$ variants $\{x_{ij}\}_{j=1}^{m_i}$ with distribution $\tilde{\mathcal{P}}_i$ and bias $\delta_i := \mathbb{E}_{\tilde{\mathcal{P}}_i}[s(x)]$.*

*Define the plug-in estimator $\hat{\theta}_i := \frac{1}{m_i} \sum_{j=1}^{m_i} \phi_{ij}$. Then:*

*(i) As $m_i \to \infty$, $\quad \hat{\theta}_i \xrightarrow{a.s.} \theta_i + \delta_i$.*

*(ii) $|\delta_i| < |\delta_i^{(0)}| \quad and \quad \mathbb{E}\left[(\hat{\theta}_i - \theta_i)^2\right] < \mathbb{E}\left[(\phi_i^{(0)} - \theta_i)^2\right]$ if $dist(\tilde{\mathcal{P}}_i, \mathcal{P}_i) < dist(\mathcal{P}_i^{(0)}, \mathcal{P}_i)$.*

*Proof.* (i) Write

$$\hat{\theta}_i - \theta_i = \frac{1}{m_i} \sum_{j=1}^{m_i} s(x_{ij}) + \frac{1}{m_i} \sum_{j=1}^{m_i} \epsilon_{ij}.$$

By the Strong Law of Large Numbers,

$$\frac{1}{m_i} \sum_{j=1}^{m_i} s(x_{ij}) \xrightarrow{a.s.} \mathbb{E}_{\tilde{\mathcal{P}}_i}[s(x)] = \delta_i,$$

and the noise term converges almost surely to zero. Hence $\hat{\theta}_i \to \theta_i + \delta_i$.

(ii) If $\tilde{\mathcal{P}}_i$ is closer to $\mathcal{P}_i$ than $\mathcal{P}_i^{(0)}$, then by stability of expectations under Wasserstein or TV convergence,

$$|\mathbb{E}_{\tilde{\mathcal{P}}_i}[s(x)] - \mathbb{E}_{\mathcal{P}_i}[s(x)]| < |\mathbb{E}_{\mathcal{P}_i^{(0)}}[s(x)] - \mathbb{E}_{\mathcal{P}}[s(x)]|.$$

Since Assumption 4.1 implies $\mathbb{E}_{\mathcal{P}_i}[s(x)] = 0$, we obtain $|\delta_i| < |\delta_i^{(0)}|$. Bias reduction follows immediately; variance shrinks because averaging over $m_i$ perturbations reduces the contribution of both $s(x)$ and the noise term. $\qquad\square$

## C. Expanding the Phrasing Space Changes Inferences

We view perturbations not as a way to recover ground truth (which is unknowable), but as a *robustness probe*: enlarging the phrasing set offers one principled way to test whether benchmark conclusions are stable to natural linguistic variation. Prior work on perturbations (Mizrahi et al., 2024; Sclar et al., 2023; Zheng et al., 2023) already illustrates that model predictions can vary substantially across alternative phrasings, but here we re-frame the problem as *inconsistencies in inference*.

For each question $i$, let $\phi_i^{\text{orig}}$ denote performance on the original phrasing and $\bar{\phi}_i = \frac{1}{m_i} \sum_{j=1}^{m_i} \phi_{i,j}$ be mean performance across $m_i$ natural perturbations. We examine the discrepancy $D_i = \phi_i^{\text{orig}} - \bar{\phi}_i$, which quantifies how much the original phrasing deviates from the performance obtained over a broader phrasing set.

*Table 3.* Description of each benchmark task, along with the number of datapoints.

| Task | Description | Number of datapoints |
|---|---|---|
| **LMEntry** | | |
| Any word from category | Yes or no question – decide if any of the five words belong to a given category | 3,000 (randomly sampled 500) |
| First alphabetically | One of two words – decide which comes first alphabetically | 3,000 (randomly sampled 500) |
| More letters | One of two words – decide which one has more letters | 3,000 (randomly sampled 500) |
| Rhyming word | One of two words – decide which one rhymes with a given word | 3,000 (randomly sampled 500) |
| **BBH** | | |
| Causal Judgment | Yes or no question – decide if the statement is causal in nature | 146 |
| Movie Recommendation | Given four movies, decide which of the four options is most similar. | 250 |
| Formal Fallacies | Determine whether or not the argument is deductively valid. | 250 |
| Snarks | Determine which of the two statements is sarcastic. | 178 |
| GPQA | Decide which of the four options is factually correct. | 448 |

## C.1. Experiments on smaller, open-source models

### C.1.1. SETUP

**Datasets.** We test on two benchmarks, Big-Bench Hard (BBH) (Suzgun et al., 2023) and LMEntry (Efrat et al., 2023), both of which have perturbed versions from (Mizrahi et al., 2024). Each dataset contains sub-tasks testing different concepts, and we use four from each category. For LMEntry, we use `any word from category` (**AWFC**), `first alphabetically` (**FA**), `more letters` (**ML**), and `rhyming word` (**RW**). For BBH, we use `causal judgment` (**CJ**), `movie recommendation` (**MR**), `formal fallacies` (**FF**), and `snarks` (**S**). See Tables 3 and 4 for details of each task.

*Table 4.* Sample question and perturbation for each of the benchmark datasets tested.

| Task | Sample |
|---|---|
| **LMEntry** | |
| All word from category | **Sample Question:** Q: Are all of the words "peach", "couch", "coat", "truck", and "shirt" types of animals? Answer either "yes" or "no". A: 
 **Sample Perturbation:** Is "animal" represented by all of the words "peach", "couch", "coat", "truck", and "shirt"? Please respond with a "yes" or "no". |
| First alphabetically | **Sample Question:** Q: In an alphabetical order, which of the words "beach" and "silver" comes first? A: 
 **Sample Perturbation:** Which word precedes the other in alphabetical order, "silver" or "beach"? |
| More letters | **Sample Question:** Q: Which word has more letters, "fit" or "rice"? A: 
 **Sample Perturbation:** Which of the two words, "rice" or "fit", is longer? |

**Table 4 – continued from previous page**

| Task | Sample |
|------|--------|
| Rhyming word | **Sample Question:** Q: Which word rhymes with the word "declare", "beer" or "wear"? A:
**Sample Perturbation:** Which word, "bear" or "wear", rhymes with the word "declare"? |

| **BBH** | |
|------|--------|
| Causal Judgment | **Sample Question:** How would a typical person answer each of the following questions about causation?
Brown is playing a simple game of dice. The game requires that Brown roll a six to win. So, hoping to get a six, Brown throws a die onto the table. Unluckily for the other players, the die lands six-up and Brown wins the game. Did Brown intentionally roll a six?
Options:
- Yes
- No
**Sample Perturbation:** Given a question about causation, classify whether a typical person would answer with "Yes" or "No".
Question: Brown is playing a simple game of dice. The game requires that Brown roll a six to win. So, hoping to get a six, Brown throws a die onto the table. Unluckily for the other players, the die lands six-up and Brown wins the game. Did Brown intentionally roll a six?
Answer: |
| Movie Recommendation | **Sample Question:** Find a movie similar to Batman, The Mask, The Fugitive, and Pretty Woman:
Options:
(A) The Front Page
(B) Maelstrom
(C) The Lion King
(D) Lamerica
**Sample Perturbation:** Please suggest a movie that is similar to Batman, The Mask, The Fugitive, and Pretty Woman. You can choose from the following options:
(A) The Lion King
(B) Lamerica
(C) The Front Page
(D) Maelstrom |
| Formal Fallacies | **Sample Question:** Here comes a perfectly valid argument: First, being a cousin of Chris is sufficient for not being a son of Kermit. We may conclude that whoever is not a son of Kermit is a cousin of Chris.
Is the argument, given the explicitly stated premises, deductively valid or invalid?
Options:
- valid
- invalid
**Sample Perturbation:** Q: Is the argument, given the explicitly stated premises, deductively valid or invalid?
First, being a cousin of Chris is sufficient for not being a son of Kermit. We may conclude that whoever is not a son of Kermit is a cousin of Chris. |

**Table 4 – continued from previous page**

| Task | Sample |
|------|--------|
| Snarks | **Sample Question:** Which statement is sarcastic?
Options:
(A) Hey, just be happy then you won't be depressed anymore
(B) Hey, just be happy that you won't be depressed anymore
**Sample Perturbation:** Which of the following sentences is sarcastic?
Options:
(A) Hey, just be happy then you won't be depressed anymore
(B) Hey, just be happy that you won't be depressed anymore
Answer: |
| **GPQA** | |
| GPQA | **Sample Question:** In a parallel universe where a magnet can have an isolated North or South pole, Maxwell's equations look different. But, specifically, which of those equations are different?
Options:
(A) The ones related to the circulation of the electric field and the divergence of the magnetic field.
(B) The ones related to the divergence and the curl of the magnetic field.
(C) The one related to the divergence of the magnetic field.
(D) The one related to the circulation of the magnetic field and the flux of the electric field.
**Sample Perturbation:** In an alternate universe where magnets can possess a lone North or South pole, how do Maxwell's equations change? Which particular equations differ in this scenario?
Options:
(A) The ones related to the circulation of the electric field and the divergence of the magnetic field.
(B) The one related to the circulation of the magnetic field and the flux of the electric field.
(C) The ones related to the divergence and the curl of the magnetic field.
(D) The one related to the divergence of the magnetic field. |

We use all questions from all benchmarks except for LMEntry, where we randomly sampled 500 questions from each task due to computational constraints. The perturbed dataset of LMEntry and BBH (Mizrahi et al., 2024) consist of different instruction formatting that was generated by both a very capable LLM and manual human labor. All prompts have been checked manually by human annotators. Each task (from both LMEntry and BBH) has around 150 distinct formatting perturbations, from which we randomly sampled 20 for each question, again due to computational constraints. Apart from the formatting perturbations, we also perturbed multiple choice order when relevant.

**Models.** We test 7 open-source autoregressive language models from three model families: `Llama-3.2` (1B and 3B parameters), `Qwen-2.5` (1.5B, 3B, 7B parameters), and `gemma` (1B, 4B params). All models were instruction-tuned. For each model, we use a temperature of 0.9. For each question, we sample 20 times to account for output stochasticity. Then, we parse each question using soft regex rules tailored to the question type. Note that we do not use LLM as a judge because the questions all present multiple answer choices, and thus answers were relatively easy to extract from raw outputs. We obtain $\theta_i$ for each question $i$ by averaging correctness over the 20 queries. The same procedure also applies to the original benchmark data, with the exception that each question was not associated with 20 different perturbations. All experiments were done using one NVIDIA L40 GPU.

### C.1.2. RESULTS

Figure 4 shows that $D_i$ frequently departs from zero, sometimes by substantial margins (up to $\pm 15$ p.p.). Importantly, the direction of deviation varies by model and task: for example, **RW** shows negative deviation for Llama-3B but positive

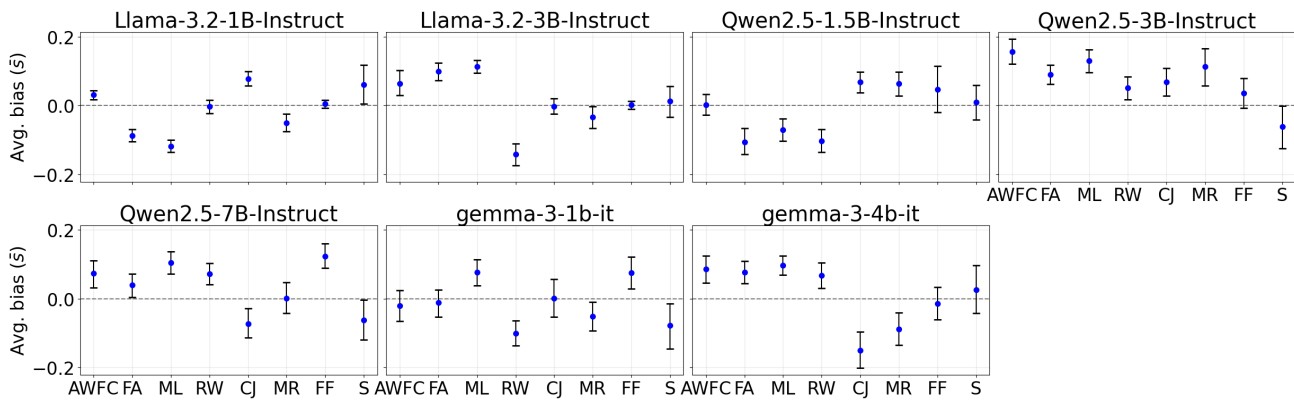

*Figure 4.* Systematic bias between estimates of accuracy based on the original benchmark data and estimate of true performance accounting for natural perturbations, across eight benchmark tasks and tested on seven LLMs.

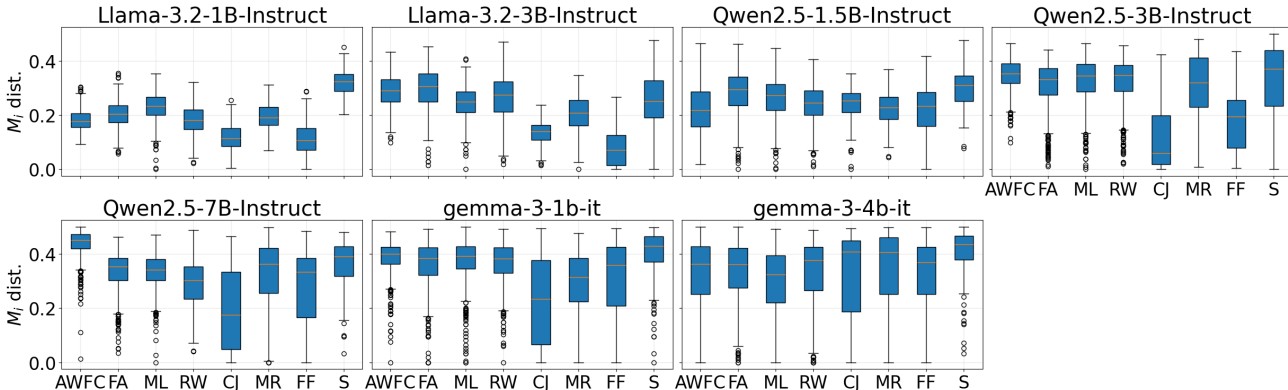

*Figure 5.* Mean absolute distance $M$, quantifying the expected deviation in performance for a new question/prompt from the benchmark distribution. Results are over all eight benchmark tasks, tested on seven LLMs.

deviation for Qwen-7B and Gemma-4B. This variability echoes prior findings (Mizrahi et al., 2024; Sclar et al., 2023) and suggests that conclusions drawn from a single phrasing are not always stable.

Figure 5 reports the empirical distribution of $M_i = \frac{1}{m_i} \sum_{j=1}^{m_i} |\phi_{i,j} - \bar{\phi}_i|$ a measure of per-item sensitivity. Across models and tasks, $M_i$ spans 10–50 p.p., indicating substantial variation in performance across natural phrasings.

Given this systematic bias, leaderboard rankings can be distorted as seen in Figure 6. First, we note that the error bars for the perturbed dataset are significantly smaller than the original dataset – this is because by averaging over 20 perturbations, we are reducing some of the internal variance in our estimate. This reduction in uncertainty allows us to better compare model performances even before considering re-rankings. For example, `Qwen-1B` and `Qwen-3.5B` are quite undifferentiated for the original **S**, we see sufficient separation between the two estimated accuracies for the perturbed **S**. Additionally, perturbed data could result in re-rankings. For example, for the original **CJ**, `Qwen-3.5B` outperforms `Qwen-7B` and `Gemma-1B` outperforms `Gemma-4B`, but the ordering is flipped for the perturbed **CJ** – which is the ordering we would have expected given the difference in model sizes.

### C.2. A preliminary probe on SoTA models

**Datasets.** We use two datasets, `movie recommendation` (**MR**) from BBH as described above, and GPQA. GPQA contains multiple-choice, graduate-level questions from the natural sciences (e.g., mathematics, chemistry). The questions are designed to be hard, in that non-experts cannot easily Google the question and retrieve the correct answer. For each question, we generate five random perturbations using `gpt-4.1-mini` using the following prompt:

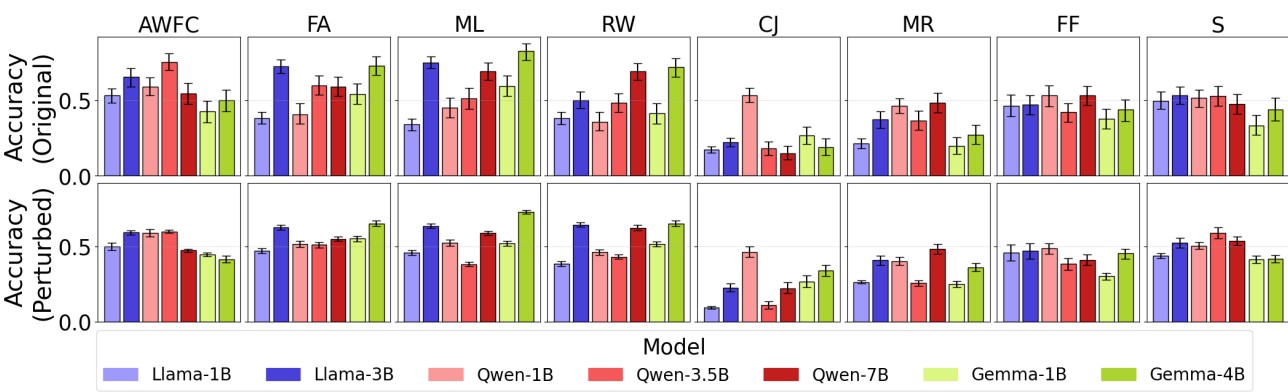

*Figure 6.* Estimated accuracies with bootstrap confidence intervals, over the original benchmark [top] and over the perturbed benchmark [bottom], for all eight benchmark tasks and over seven LLMs.

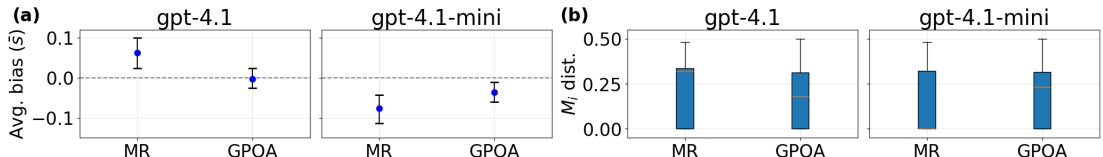

*Figure 7.* Same as Figures 4 and 5 but with `gpt-4.1` and `gpt-4.1-mini` models and only on **MR**, **GPQA** data.

```
Please generate 5 different perturbations of the prompt below, keeping all
the pertinent information but expressed in a different way.  When the prompt
gives multiple choices, DO NOT SHUFFLE THE OPTIONS, but feel free to re-word
each option.  Be as clear as possible.  Structure your response with
"Perturbation 1:  [PROMPT]
Perturbation 2:  [PROMPT]" and so on.
Prompt:
```

All of the questions were qualitatively checked to ensure that the perturbations remain clear and do not lose any pertinent information. Then, we randomly shuffle the answer choices and add them onto the question to construct the full input prompt. See Table 4 for an example.

**Models.** We test `gpt-4.1` and `gpt-4.1-mini`, state-of-the-art autoregressive language models. We use `gpt-4.1-mini` to randomly generate question perturbations on GPQA. `gpt-4.1` achieves 71% and 47% accuracy on the original benchmarks `MR` and GPQA, while `gpt-4.1-mini` achieve 65% and 47%, respectively. This suggests that GPQA is a frontier dataset, since even the best current models (save for reasoning/test-time inference) are incorrect most of the time on that benchmark. For each model, we use a temperature of 0.7. For each question, we sample 5 times to account for output stochasticity. Then, we parse each question using soft regex rules tailored to the question type, and recover $\theta_i$ by averaging correctness for each query. All experiments were done through API calls to OpenAI.[2]

**Results.** Figure 7 shows analogous results for `gpt-4.1` and `gpt-4.1-mini` on **MR** and GPQA. Although deviations are generally smaller (roughly 0–8 p.p. for **MR** and somewhat larger for GPQA), they are non-negligible. Since these experiments span only a handful of tasks, we treat them as *initial probes*. Nonetheless, they indicate that even state-of-the-art models can exhibit notable phrasing sensitivity on harder, frontier problems.

## D. Inference Methods

In this section, we provide detailed algorithms for the four methods described in Section 4.2.

---

[2]See https://openai.com/api/.

---

**Algorithm 1** Clustered Bootstrap for Estimating Accuracy (CBA)

---

**Require:** Observed scores $\{\phi_{ij}\}$ for $i = 1, \ldots, n$, prompts per item $\{m_i\}$, resamples $B$
 1: Initialize empty vector $\mathcal{T} \leftarrow []$
 2: **for** $b \leftarrow 1$ to $B$ **do**
 3:     Sample with replacement $n$ indices $\mathcal{I}^{(b)} \subseteq \{1, \ldots, n\}$ {resample items}
 4:     **for all** $i \in \mathcal{I}^{(b)}$ **do**
 5:         $\hat{\theta}_i^{(b)} \leftarrow \frac{1}{m_i} \sum_{j=1}^{m_i} \phi_{ij}$
 6:     **end for**
 7:     $\hat{\theta}^{(b)} \leftarrow \frac{1}{n} \sum_{i \in \mathcal{I}^{(b)}} \hat{\theta}_i^{(b)}$
 8:     Append $\hat{\theta}^{(b)}$ to $\mathcal{T}$
 9: **end for**
10: **return** Empirical distribution $\mathcal{T}$ (use percentiles for $(1 - \alpha)$ CI)

---

### D.1. Inference under CTT

Under classical test theory (CTT), each observed score $\phi_{ij}$ is modeled as a noisy realization of an underlying item-level latent accuracy $\theta_i$, with repeated prompts serving as conditionally independent measurements of the same item. The quantity of interest is the population-level mean accuracy $\theta = \mathbb{E}[\theta_i]$, which in practice is estimated by averaging item-wise means $\hat{\theta}_i = m_i^{-1} \sum_{j=1}^{m_i} \phi_{ij}$. Because prompts are nested within items, naïvely bootstrapping individual observations would underestimate uncertainty by ignoring within-item dependence.

Algorithm 1 implements a clustered bootstrap that respects this hierarchical structure. Each bootstrap replicate resamples items (clusters) with replacement, recomputes the corresponding item-level accuracies, and aggregates them to form a replicate estimate of the overall accuracy. The resulting empirical distribution of $\hat{\theta}^{(b)}$ approximates the sampling distribution of the estimator under the CTT model, allowing percentile-based confidence intervals that are robust to heteroskedasticity in the number of prompts per item and arbitrary within-item dependence.

### D.2. Inference under IRT

Assume we have an ecosystem of $K$ generative models. We will use Laplace-approximated marginal maximum likelihood estimation, which places priors on parameters $\theta$, integrates them out by taking their mode (using Laplace approximation), then directly maximizes the marginal likelihood over the remaining parameters. Note that any valid inference method will work, and there are well-established methods in the literature; for example, Thissen & Steinberg (2020) outline an expectation-maximization (EM) algorithm and a Markov chain Monte Carlo algorithm based on Bayesian inference. Note that these parameters are **identifiable** as long as either $\theta_k$ or $b_i$ are anchored (Baker & Kim, 2004), hence why we anchor $\theta_k$ to be mean zero and variance one through our prior $\theta_k \sim \mathcal{N}(0, 1)$.

**Preliminaries.** Recall that

$$\phi_{ijk} \sim \text{Bernoulli}(\bar{P}_{ik}(\theta)), \quad \bar{P}_{ik}(\theta) = \sigma(a_i(\theta_k - b_i)),$$

where we re-introduce the index $k$ because we have multiple models now). We therefore have

$$Y_{ik} = \sum_{j=1}^{m} \phi_{ijk} \sim \text{Bin}(m, \bar{P}_{ik}(\theta)).$$

**Log-Likelihood.** We now derive the log-likelihood for which we optimize. Let $\xi = \{a_1, \ldots, a_n, b_1, \ldots, b_n, \theta_1, \ldots, \theta_K\}$ capture the set of parameters we want to infer. By Bayes rule, $p(\xi|Y) \propto p(Y|\xi)p(\xi)$ and thus

$$\log p(\xi|Y) \propto \log p(Y|\xi) + \log p(\xi).$$

Since $Y$ follows a binomial distribution,

$$\log p(Y|\xi) = \sum_{i=1}^{n} \sum_{k=1}^{K} \left[ Y_{ik} \log \bar{P}_{ik} + (m - Y_{ik}) \log(1 - \bar{P}_{ik}) \right].$$

For the prior likelihood, we a prior on $\theta_k$ so that parameters are identifiable. In particular, $\theta_k \sim \mathcal{N}(0, 1)$, which yields

$$\log p(\theta) = -0.5 \sum_k \theta_k^2.$$

In sum, we have the following posterior log likelihood:

$$\ell_{\text{post}} = \sum_{i,k} \left[ Y_{ik} \log \bar{P}_{ik} + (m - Y_{ik}) \log(1 - \bar{P}_{ik}) \right] - \frac{1}{2} \sum_k \theta_k^2. \tag{7}$$

We then minimize $-\ell_{\text{post}}$ using L-BFGS-B[3] over $[\log a_i, b_i, \theta_k]$, and taking the exponentials over $\log a_i$ to enforce positivity. We summarize the procedure below:

1. Construct $\ell_{\text{post}}$ on parameter vector $\tilde{\xi} = [\log a_i, b_i, \theta_k]$.

2. Optimize $-\ell_{\text{post}}$ using L-BFGS-B

3. Recover $\xi$ by taking $\hat{a}_i = \exp(\hat{\log a_i})$.

---

**Algorithm 2** Latent Ability Adaptive Test (LAAT)

---

**Require:** Item params $\{(a_i, b_i)\}_{i=1}^N$, prior $\theta \sim \mathcal{N}(\mu_0, \sigma_0^2)$, perturbations $m$, stopping criterion $C$
1: $\theta \leftarrow \mu_0, \mathcal{I} \leftarrow \frac{1}{\sigma_0^2}, \mathcal{A} \leftarrow \emptyset$
2: **while** $C$ not met **do**
3:     Select $i^* = \arg \max_{i \notin \mathcal{A}} I_i(\theta)$, where

$$I_i(\theta) = m a_i^2 \bar{P}_i(\theta) \left(1 - \bar{P}_i(\theta)\right), \quad \bar{P}_i(\theta) = \sigma(a_i(\theta - b_i))$$

4:     $\mathcal{A} \leftarrow \mathcal{A} \cup \{i^*\}$
5:     Query model on $m$ perturbations of $i^*$, get $\{\phi_{i^* j}\}_{j=1}^m$
6:     $\phi_{i^*} \leftarrow \frac{1}{m} \sum_{j=1}^m \phi_{i^* j}$
7:     $S \leftarrow \sum_{i \in \mathcal{A}} m a_i \left(\phi_i - \bar{P}_i(\theta)\right)$
8:     $\mathcal{I} \leftarrow \sum_{i \in \mathcal{A}} m a_i^2 \bar{P}_i(\theta) \left(1 - \bar{P}_i(\theta)\right)$
9:     Update $\theta \leftarrow \theta + \frac{S}{\mathcal{I}}$ {Newton–Raphson/Fisher scoring}
10: **end while**
11: **return** $\theta$ (ability estimate), $1/\sqrt{\mathcal{I}}$ (SE), items asked $\mathcal{A}$

---

### D.3. Inference under CDM

Under a cognitive diagnostic model (CDM), observed responses are generated by an interaction between model-level latent skills $\theta_k$ and item-level attributes encoded by a $Q$-matrix. Each item $i$ is associated with parameters $\psi_i$ governing the link function $f_{\text{CDM}}(\theta, Q_i; \psi_i)$, which maps latent skills and required attributes to a distribution over observed scores. Repeated perturbations for the same item are treated as noisy measurements and are first aggregated to obtain item-level responses $\bar{\phi}_{ki}$ for each model.

Algorithm 3 performs joint estimation of latent skills and item parameters by alternating optimization. Conditional on current item parameters, model-specific skills are updated by maximizing their posterior likelihood given all item responses, using second-order or quasi-Newton updates. Conditional on the updated skills, item parameters are refined by maximizing the expected complete-data log-likelihood under the current skill posteriors, with optional regularization through priors on $\psi$. This alternating procedure is iterated until convergence, yielding point estimates of both skills and item characteristics.

Uncertainty in the estimated skills can be approximated using local curvature (e.g., the inverse Hessian of the log-posterior) or via a clustered bootstrap over items. The resulting estimates and intervals capture both measurement noise from repeated perturbations and structural uncertainty induced by the item–skill interaction specified by the CDM.

---

[3]See https://docs.scipy.org/doc/scipy/reference/optimize.minimize-lbfgsb.html.

---

**Algorithm 3** CDM: Joint Item and Skill estimation

---

**Require:** $Q$-matrix, CDM link $f_{\text{CDM}}(\theta, Q_i; \psi_i)$ with item params $\psi_i$, perturbations $\{x_{ij}\}$, responses $\{\phi_{kij}\}$ for models $k$
1: Aggregate perturbations: $\bar{\phi}_{ki} \leftarrow \frac{1}{m_i} \sum_j \phi_{kij}$
2: Initialize item parameters $\psi = \{\psi_i\}$ and skills $\{\theta_k\}$
3: **repeat**
4:     *// Skill (ability) updates*
5:     **for** each model $k$ **do**
6:         Form log-posterior $\log p(\theta_k) + \sum_i \log p(\bar{\phi}_{ki} \mid \theta_k, Q_i, \psi_i)$
7:         Update $\theta_k \leftarrow \theta_k + \mathcal{I}(\theta_k)^{-1} S(\theta_k)$ {Newton / L-BFGS on log-posterior}
8:     **end for**
9:     *// Item-parameter updates*
10:    Update $\psi$ to maximize $\sum_{k,i} \mathbb{E}_{p(\theta_k)}[\log p(\bar{\phi}_{ki} \mid \theta_k, Q_i, \psi_i)] + \log p(\psi)$ (e.g. gradient or L-BFGS step)
11: **until** convergence of $\{\theta_k\}$ and $\psi$
12: Approximate skill-level uncertainty via inverse Hessian or item bootstrap
13: **return** $\{\hat{\theta}_k\}$ and associated skill posteriors / intervals

---

## D.4. Inference under BNSM

Under the Bayesian network skill model (BNSM), latent skills are represented as nodes in a probabilistic graphical model with a structured dependency graph. A multivariate Gaussian prior $p(S; \mu, \Sigma)$ captures both marginal skill uncertainty and dependencies between skills, while each item $i$ is associated with a conditional probability distribution (CPD) $p(\phi_i \mid S; w_i)$ that links skill mastery to observed responses through a logistic likelihood. As in previous settings, repeated perturbations of the same item are treated as noisy measurements and are aggregated to form item-level evidence $\bar{\phi}_{ki}$ for each model.

Algorithm 4 alternates between approximate posterior inference over latent skills and parameter estimation. Given current model parameters, posterior distributions over skills are computed for each model using approximate Bayesian network inference methods, such as variational inference, Laplace approximations, or belief propagation for mixed continuous–discrete graphs. Conditioned on these approximate posteriors, the prior parameters and item-level CPDs are updated by maximizing the expected complete-data log-likelihood. This procedure iterates until joint convergence of parameters and posteriors.

The resulting posterior distributions $q_k(S)$ provide skill-level mastery estimates along with coherent uncertainty quantification through marginal credible intervals. Additional uncertainty arising from finite item sampling can be assessed via an optional clustered bootstrap over items, yielding inference that reflects both epistemic uncertainty in skill estimation and measurement noise in observed responses.

---

**Algorithm 4** BNSM: Posterior Inference

---

**Require:** BN structure over skills $S$, Gaussian prior $p(S; \mu, \Sigma)$, logistic item CPDs $p(\phi_i \mid S; w_i)$, perturbations $\{x_{ij}\}$, responses $\{\phi_{kij}\}$ for models $k$
1: Aggregate evidence: $\bar{\phi}_{ki} \leftarrow \frac{1}{m_i} \sum_j \phi_{kij}$
2: Initialize Gaussian prior params $(\mu, \Sigma)$ and item CPD weights $w = \{w_i\}$
3: **repeat**
4:     *// Inference over skills given current parameters*
5:     **for** each model $k$ **do**
6:         Run approximate BN inference (e.g. continuous–discrete belief propagation / variational / Laplace) to obtain $q_k(S) \approx p(S \mid \{\bar{\phi}_{ki}\}_i, \mu, \Sigma, w)$
7:     **end for**
8:     *// Parameter updates given current posteriors*
9:     Update $(\mu, \Sigma, w)$ to maximize the expected complete log-likelihood $\sum_k \mathbb{E}_{q_k(S)}[\log p(S; \mu, \Sigma) + \sum_i \log p(\bar{\phi}_{ki} \mid S; w_i)]$
10: **until** convergence of parameters and posteriors
11: For each model $k$, extract posterior mastery summaries (e.g. marginal means $\mathbb{E}_{q_k}[S_\ell]$ and credible intervals)
12: Optionally bootstrap items to quantify additional uncertainty
13: **return** Posterior skill mastery $\{q_k(S)\}$ and summaries per skill

---

## D.5. Optimal Sampling under Budget Constraint

Given budget $B$, we can optimize the number of samples $m_i$ for each question $i$ using Neyman allocation (Neyman, 1992). Assume that $B$ is the number of times we can query the LLM. We treat each question $i$ as an equally weighted stratum.

Given the standard deviation in performance of each stratum $\sigma_i$ (i.e., how much performance deviates across perturbations of the same question), the optimal sample size is given by:

$$m_i = \frac{\sigma_i B}{\sum_{j=1}^{n} \sigma_j}.$$

However, $\sigma_i$ is unknown a priori, so we need to first estimate it. We propose a two-step procedure in Algorithm 5.

---

**Algorithm 5** Two-Step Neyman Allocation Procedure

---

**Require:** Budget $B$, number of questions $n$, initial sample size $m_0$
**Ensure:** Allocation of total samples $m_i = m_i^{(0)} + m_i^{(1)}$ for each question $i$
  1: **[Step 1] Initial Sampling:**
  2: **for** each question $i = 1$ to $n$ **do**
  3:     Query the LLM $m_0$ times for question $i$ to obtain responses $\{\phi_{ij}\}_{j=1}^{m_0}$
  4:     Compute empirical mean: $\hat{\theta}_i = \frac{1}{m_0} \sum_{j=1}^{m_0} \phi_{ij}$
  5:     Estimate standard deviation: $\hat{\sigma}_i = \sqrt{\frac{1}{m_0-1} \sum_{j=1}^{m_0} (\phi_{ij} - \hat{\theta}_i)^2}$
  6: **end for**
  7: Compute remaining budget: $B' = B - n \cdot m_0$
  8: **[Step 2] Neyman Allocation:**
  9: **for** each question $i = 1$ to $n$ **do**
 10:     Allocate additional samples: $m_i^{(1)} = \left\lfloor \frac{\hat{\sigma}_i B'}{\sum_{j=1}^{n} \hat{\sigma}_j} \right\rfloor$
 11:     Total samples for question $i$: $m_i = m_0 + m_i^{(1)}$
 12: **end for**
 13: **return** $\{m_i\}_{i=1}^{n}, \{\phi_{ij}\}_{j=1}^{m_0}$

---

### D.5.1. TRADE-OFF BETWEEN $n$, $m_i$ AND REPEATED SAMPLING

So far, we've discussed given a fixed $n$, how to choose $m_i$ for each question $i$. For some large datasets, however, $n$ is too large given one's budget. In this case, we recommend choosing some small $m \geq 3$ for each question to get a sufficient estimate of variance, and backing out how much $n$ one can afford with $B$ queries. This is because in order for confidence intervals to be accurate, we need enough $m$ to simulate "independent" draws of question phrasings. However, we need sufficient $n$ since it is the most influential in reducing variance.

Note that **we do not** advocate for repeated sampling of outcomes for every perturbation to reduce the variance of $\epsilon_i$. This is because we are sampling independently, and that variance term will wash out as $n$ becomes large.

## E. Inference Experiment Details

We provide details on our implementation of Algorithm 2 in our experiments. In particular, we used the following stopping criterion: stop when the difference in standard errors between the most recent iteration and the previous one is less than 0.0001. This stopping condition allows us to stop inference until we have squeezed out as much information as possible from the given questions; any additional questions will only improve our estimates marginally.

Figures 8 and 9 show inferences of accuracy and ability based on Algorithms 1 and 2, respectively, on all eight benchmark tasks and over seven open-source LLMs. The full suite of experiments for Algorithm 3 is displayed in Figure 3 since they are aggregated into five latent skills, but Figure 10 show results for all skills for BNSM.

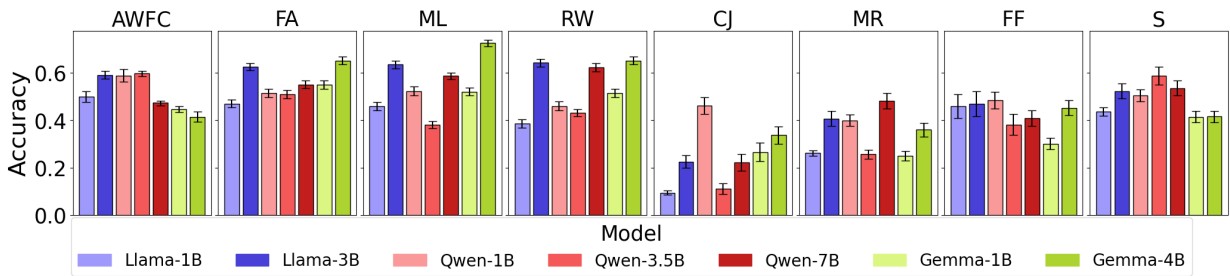

*Figure 8.* Estimates of accuracy using CBA (Algorithm 1) on eight benchmark tasks and over seven open-source LLMs.

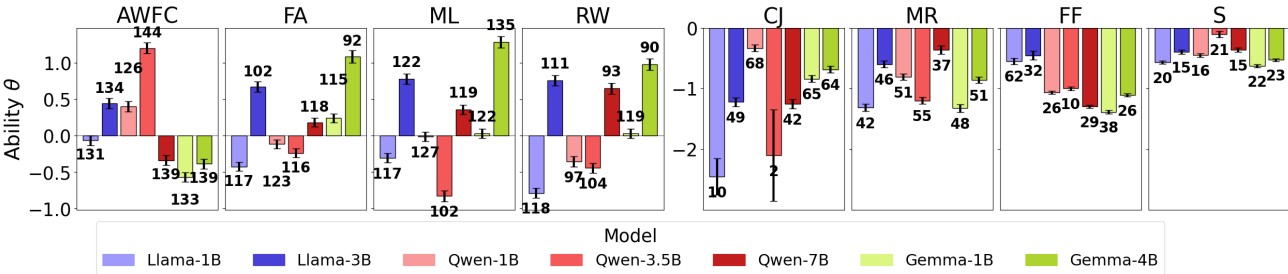

*Figure 9.* Estimates of ability using LAAT (Algorithm 2) on eight benchmark tasks and over seven open-source LLMs. Numbers in bold indicate number of questions asked in the adaptive test. Each question is associated with 20 random perturbations.

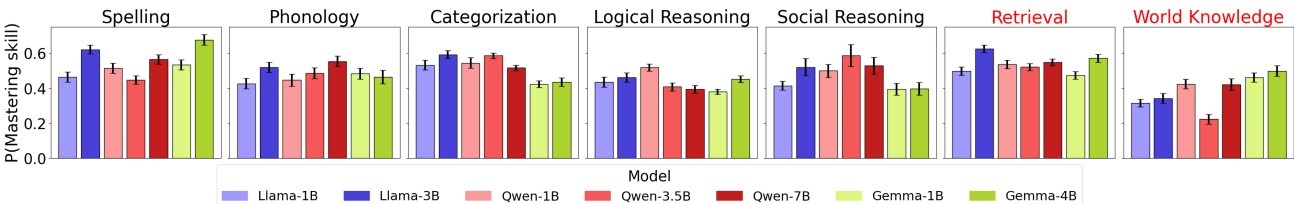

*Figure 10.* Estimates of skill mastery using (Algorithm 4) on eight benchmark tasks and over seven open-source LLMs.

