# OpenReview forum: "Position: AI Evaluations Should be Grounded on a Theory of Capability"
_ICML.cc/2026/Position_Paper_Track — ICML 2026 Position Paper Track regular_

### Official Review · Reviewer_hxgF · 2026-03-08

**Significance:** 4
**Argument Clarity:** 2
**Rating:** 5
**Confidence:** 3

**Questions:**

I believe this is generally a good position paper that is likely to lead to interesting discussions within the (very active) evals community. Because of this, I am currently leaning towards accepting this paper as its position is clear and well-supported. That being said, I would like to state that (1) my background is not in psychometrics, so I may be unaware of key works that were not discussed in this paper, and (2) I still have some concerns about the clarity of some portions of this paper.  The following questions, however, may help clarify some of the concerns I raised above and could improve the quality of this work if they are answered in the updated manuscript:

1. **[Critical, Related Work Clarification]** Could you please comment on how your position aligns or disagrees with that posed by the work of Zhuang et al. [1]? Given that this is also a position paper arguing for the use of psychometrics in AI evals in ICML 2025, I believe it is important to compare and contrast your argument with theirs and, at the very least, include this work as part of your related work section.
2. **[Major, Clarification]** Consider the scenario where a model performs very well on downstream tasks based on traditional average performance metrics (e.g., average accuracy), but its inferred performance on individual skills when using a pre-determined skill structure is low. How would one discriminate between (1) the case where evaluation is wrong because the evaluators chose the wrong skill structures or modeling assumptions for noise and (2) a setup where the model is, say, "right for the wrong reasons", remaining accurate because it exploits spurious correlations (so the actual low skill performance is an accurate representation of what is going on)?
3. **[Major, Clarity]** Can you please clarify how the different skills for CDM and BNSM are estimated? More specifically, how do you ensure that your estimates are **aligned** to what we would expect each skill to represent (e.g., a score for “spelling” really represents the model’s ability to spell)? Currently, many of these key details are pushed to an appendix, and it appears that alignment is not explicitly verified anywhere but instead assumed to emerge from the skill-to-task structure alone (please correct me if I misinterpreted this). Regardless, I would strongly suggest that authors explicitly discuss the (potential) problem of alignment in scores in the main body of the paper. This is particularly important as misalignment between skill scores and actual skill performance can be a key limitation and barrier for using theories of capability that depend on skill structures.
4. **[Minor, Impact]** Related to the questions above, in practice, how would you expect practitioners to design “good” skill structures when using CDM/BNSM-like theories of capabilities during evaluation? As an example, even though there is a link between "Spelling" and "ML" in Figure 2, one can likely argue that "spelling" is not really required for "more letters" (e.g., I could tell you whether one sequence of symbols is longer than another without having any understanding of what each symbol represents).


---

### References

[1] Zhuang, Yan, et al. "Position: AI evaluation should learn from how we test humans." *Forty-second International Conference on Machine Learning Position Paper Track*. 2025.

**Alternative Views Section:**

Yes

**Compliance With Llm Reviewing Policy A Conservative:**

Affirmed.

**Discussion Potential:**

4

**Final Justification:**

Please refer to my questions section for my justification as to why I believe this paper should be accepted. Moreover, as indicated in my rebuttal comment, the authors addressed all the concerns and questions I raised in that section. As such, I stand by my original assessment and score for this paper.

**Paper Summary:**

This paper argues that evaluating complex AI systems with single aggregate metrics (e.g., mean accuracy across utility tasks) does not provide a complete picture of how these models perform or what capabilities they have. As such, the authors argue that AI evaluations should draw inspiration from psychometrics, where the question of providing a complete yet unbiased assessment of an agent (e.g., a test taker) is central. In particular, this work argues that by evaluating AI systems under different theories of capabilities such as Classical Test Theory, Item Response Theory, Cognitive Diagnostic Models, and Bayesian Network Skill Models, we can get a more complete picture of the strengths and weakenesses of AI systems while making any assumptions taken during evaluation explicitly clear (something that is entirely lost when we simply look at aggregate single-value metrics like average accuracy). This position is then supported by a proof-of-concept demonstrating how different theories of capability can be used to evaluate existing LLMs across several downstream tasks, revealing differences across models and tasks that are not exposed by traditional evaluation pipelines.

**Position:**

Yes

**Position In Title:**

Yes

**Related Work:**

2

**Strengths And Weaknesses:**

### Strengths

Thank you for submitting this very interesting position paper! After reading this work, I believe its main strengths are the following:

1. **[Critical, Quality of Position]** The position is clearly presented in this paper (including in the title, abstract, and instructions using clear visual markers). Moreover, the position is supported by connecting AI evals (e.g., LLM reasoning tasks) to the rich body of psychometric literature that has addressed similar questions for a long time. This is done by introducing several common theories of capabilities in psychometrics and then arguing that these theories provide a more nuanced and complete view of a test-taker's (e.g., an LLM) capabilities when their assumptions are clearly stated rather than implicit. The examples this paper provides in Section 5 make its position even clearer and more grounded, showing how an explicit theory of capability can really affect how we interpret eval results.
2. **[Critical, Significance]** Given the relevance and popularity of evaluations in ML/AI, particularly for those working with LLMs, foundation models, and LLMs, I believe this position paper is likely to be of interest to the ICML community.
3. **[Major, Discussion Impact]** Given how common single-metric aggregate benchmarks are in AI, I believe any paper arguing for a shift from that will have the potential to generate a discussion in the community. Therefore, I believe this work’s position is likely to generate discourse and can potentially lead to changes in how future benchmarks are designed.
4. **[Minor, Related Work]** This paper properly links its contributions and positions with previous works both in model evaluation and in psychometrics, particularly in Sections 2 to 4. Nevertheless, as discussed below, I would’ve preferred if the extended literature discussion in the appendix had been included as part of the main body of the paper. Moreover, a clear omission from this work is the previous ICML position paper by Zhang et al. [1] arguing for the inclusion of psychometric-like evaluations for modern AI systems.
5. **[Major, Clarity]** The paper is very well-written, organized, and easy to follow. In particular, I found almost no typos, and the paper’s structure makes it easy for readers to follow a cohesive argument throughout.

### Weaknesses

In contrast, I believe the following are key weaknesses of this work (sorted in terms of importance):

1. **[Critical, Proper Framing]** Some highly relevant/related previous position papers at ICML, such as that by Zhuang et al. [1], are not discussed in this work, making it difficult for readers to understand the alignment or differences between that work’s position and this paper's position.
2. **[Major, Related Work]** Related to the point above, I would strongly discourage the authors from putting the related work in an appendix, as, in my opinion, this is generally not good practice (meaning important and relevant works may not be properly seen by readers). Therefore, if accepted, I would suggest adding a proper related work section to the main paper.
3. **[Major, Clarity]** Some important details on the proof-of-concept in Section 5, which would be very helpful to really understand how to use some of the frameworks advocated by this paper’s position, are unfortunately buried in the appendix. This makes some of the arguments in this work less concrete and harder to situate in practical settings.

### Potential Typos and Miscellaneous Suggestions

In case this is of any help, I also compiled the following potential typos and miscellaneous feedback while reading this work:

- **[Major, Writing Flow]** In line 67, col 2, assumption 2.1 is discussed as if it was already introduced before (making that paragraph a bit hard to follow at first). I would suggest rewriting this to first indicate the origin/need for a new assumption.
- **[Major, Missing Citation]** When discussing the Generalized DINA model, you should cite the original work proposing it by De La Torre [2].
- **[Major, Structure]** I would suggest adding the appendices in the same file as the main document, as otherwise the many links to the appendices in the main body are broken.
- **[Minor, Nit]** In Figure 1(ii), I would suggest redrawing the graph so that edges do not overlap with parameter nodes (and that edges from parameters nodes do not overlap with edges from abilities to questions). A similar comment applies to Figure 2.

---

### References

[1] Zhuang, Yan, et al. "Position: AI evaluation should learn from how we test humans." *Forty-second International Conference on Machine Learning Position Paper Track*. 2025.

[2] De La Torre, Jimmy. "The generalized DINA model framework." *Psychometrika* 76.2 (2011): 179-199.

**Support:**

3

---

> ### Author Rebuttal · Authors · 2026-03-30
>
> Thank you for the positive and thorough review! We appreciate that the reviewer found our position interesting and timely. Below, our response is labeled by Weaknesses (**W**) and Questions (**Q**).
>
> ## W1 and Q1
> We actually do cite a previous version of this paper in Section 3.2 (line 305), but appreciate the reviewer bringing up the most updated version. In short, this paper is advocating specifically for IRT models to be the paradigm for benchmarking, focusing on the ability to create adaptive tests for LLMs that reduce sample complexity. Our work is broader than that, because we discuss the many possible human-derived models of ability beyond IRT (e.g., CDMs are widely used in educational testing).
>
> ## W2 and W3
> We agree with the reviewer that the Related Work and details on Section 5 should be added to the main text from the Appendix. Should the paper be accepted, we will make sure to incorporate them in the camera-ready, since we will have additional space.
>
> ## Potential Typos and Miscellaneous Suggestions
> Done to all of the typos and suggestions.
>
> ## Q2
> Interesting question! Distinguishing between the two explanations would likely require external validation; otherwise, endogenously it would be unidentifiable. For example, one could design targeted questions that more cleanly isolate the skill, compare several plausible skill structures to test robustness, or analyze the traces/reasoning to see if the model is taking shortcuts. In general, our framework should be thought of as a test given some hypothesis (the theory of ability), and not a means for causal attribution. This is true in general for any hypothesis test, where the data might confirm or reject some hypothesis, but the reasons behind why must be further explored.
>
> ## Q3
> We believe there are two separate factors underlying this question: one methodological and one philosophical.
>
> *Methodological*: for CDM, we use penalized maximum a posteriori estimation (Alg 3) and for BNSM, we use a standard posterior inference method (Alg 4). We only briefly mentioned them in Section 5.2 because they are rather standard methods for these types of models. These methods are known to properly identify these skill estimates **conditional on the prespecified skill structure** (see e.g., [1, 2]). An easy way to verify this is to compare the accuracies for, say, FA and ML, and compare them to the skill estimates for “Spelling”. The relative ordering should be roughly similar.
>
> *Philosophical*: We suspect the reviewer was instead asking how we could verify that a high score in spelling actually corresponds to the model’s true ability to spell, which is about construct validity. This is a hard question that plagues benchmarking/tests in general, and is the topic of a separate literature (see e.g., [3]). Some things one can do to test construct validity is to find other assessments or benchmarks that test a similar ability (say, other spelling tasks) and measure how correlated the abilities are. This is well-established for human testing but we have not seen it being done for AI benchmarking. We will make sure to add this to the future work section in the camera-ready should the paper be accepted.
>
> ## Q4
> The short answer is that domain experts should be looped in to develop “good” skill structures, or at least have a reasonable justification for the choice. There is never ground truth in this type of task, and the best we can hope is that evaluators are transparent in these design decisions (which we consistently emphasize throughout the paper). As the reviewer mentioned, one could have an objection with our skill structure in Figure 2 with spelling not being a requisite for the “ML” task, but Fig 2 is intended to illustrate one of many reasonable structures.
>
> [1] De La Torre, Jimmy. "The generalized DINA model framework." Psychometrika 76.2 (2011).
>
> [2] Almond, Russell G. et al. “Modeling Diagnostic Assessments with Bayesian Networks.” Journal of Educational Assessments (2007).
>
> [3] Raji, Deborah et al. “AI and the Everything in the Whole Wide World Benchmark.” NeurIPS (2021).

---

> > ### Author Rebuttal · Reviewer_hxgF · 2026-04-02
> >
> > Thank you for your careful and clear replies. All of my concerns have been addressed. Because my score was already on the "acceptance" side, I will keep it as such (as my background is not in this specific field). Nevertheless, I believe this is an overall good paper, and I wish the authors the best of luck.

---

### Official Review · Reviewer_hbpx · 2026-03-12

**Significance:** 3
**Argument Clarity:** 2
**Rating:** 3
**Confidence:** 3

**Questions:**

If benchmark questions could be categorized into different skill sets, could reporting fine-grained accuracies solve the issue of current benchmarks?

**Alternative Views Section:**

Yes

**Compliance With Llm Reviewing Policy A Conservative:**

Affirmed.

**Discussion Potential:**

3

**Final Justification:**

The concerns of my reviews are addressed. However, I feel the position could be further clarified about the theory of capability and current benchmarks.

**Paper Summary:**

This paper presents the position that evaluations of a generative model should explicitly account for the model's capabilities when testing models on benchmarks.  The paper discusses existing theories of capability and explains where the current evaluation stands in relation to them. The paper presents a comparative study of proof-of-concept evaluation under capability theory estimation. An empirical study is conducted on 7 models and 2 datasets to demonstrate performance comparison across different capability theories. Finally, it discusses two alternative views related to the position.

**Position:**

Yes

**Position In Title:**

Yes

**Related Work:**

2

**Strengths And Weaknesses:**

Strengths
- The meaning of capability is well explained.
- How current models have become general-purpose, and the performance reported could be more meaningful, provides a good motivation for the evaluations to report the capability of the model being tested.

Weakness
- The paper discusses more about explaining the theory of capability rather than providing a systematic review of how the existing benchmarks are not grounded on a theory of capability, which is their position. It would be stronger if the paper discussed the failure scenarios of existing benchmarking.

**Support:**

2

---

> ### Author Rebuttal · Authors · 2026-03-30
>
> Thank you for the review!
>
> ## Weakness
>
> We only use Section 2 to explain the theory of capability. **The remainder of the paper is focused precisely on what the reviewer is suggesting.** In Section 3, we show that the vast majority of benchmarks assume a very simplified capability model – in particular, that they implicitly assume this theory of capability and do not properly state the underlying assumptions behind why this theory is reasonable. Then, in Section 5 and Appendix C.2, we show that this capability model has a “failure” in that it only reveals one aspect of the model’s capabilities, but that many others exist.
>
> ## Question
>
> Great question! The reviewer’s suggestion actually falls under the CDM and BNSM models in our paper. Our claim is not that they will “solve” the issue of current benchmarks: just that there exist these other reasonable models that could change your inferences about model capabilities.
>
> On a broader note, the reviewer's question already embeds some assumptions about capability: that models have some ability on well-defined (discrete) skill sets. This is precisely the point of our paper: evaluators should be explicit about the assumptions they make about models’ capabilities and the underlying data generating process (in the reviewer’s case, a prior about heterogeneous skills based on domain).

---

> > ### Author Rebuttal · Reviewer_hbpx · 2026-04-03
> >
> > Thank you for the rebuttal. My queries are clarified, and I have one clarifying question remaining.
> > So the position is that **although the current methods are assuming a theory of capability implicitly, they are not explict about it, making it difficult to interpret what capability they have.**, is it correct?
> >
> > If that's the case, the position stance might be sligtly modified in that direction for clarity of the paper.

---

### Official Review · Reviewer_xFze · 2026-03-13

**Significance:** 3
**Argument Clarity:** 2
**Rating:** 3
**Confidence:** 5

**Questions:**

1. This paper has shown inferences drawn from IRT, CDM, and BNSM to highlight the sensitivity of the evaluation outcome to the chosen theory. I’m courious that if evaluators are free to define the theory, how do we prevent model developer from selectively reporting the capability theory that has the most favorable capability estimates for their model?

2. The paper uses prompt perturbations as its main proof source of uncertainty. Do the authors believe the same argument would hold equally strongly for other confounders they discuss, such as inference time strategies, context length, or best-of-N sampling? If so, can they give a more concrete sketch of how the framework would extend?

3. The authors provide a qualitative trade-off discussion in Section 6, but stop short of an actionable framework. Given the immense cost of evaluations, could the authors provide a more concrete, decision-tree-style framework in their revision? Specifically, how should a practitioner mathematically or empirically weight predictive validity against sample efficiency when choosing between IRT and BNSM for LLM?

**Alternative Views Section:**

Yes

**Compliance With Llm Reviewing Policy A Conservative:**

Affirmed.

**Discussion Potential:**

2

**Final Justification:**

Thank you for the thoughtful rebuttal. I appreciate the clarifications, especially the emphasis on making theoretical commitments explicit and auditable rather than encouraging post hoc ''theory shopping''. That said, my main concern remains: the paper still does not offer a sufficiently concrete way to constrain or validate theory choice in practice, and the response to the scalability and decision-framework questions remains intentionally high level. I understand that the authors view these issues as partly normative, but for me this also limits the paper’s practical contribution beyond the core conceptual argument. Overall, I still find the paper interesting and potentially valuable for discussion, but I do not feel the rebuttal fully resolves the key concerns I raised, so I am maintaining my original score

**Paper Summary:**

This paper argues that the prevalent approach of reporting benchmark scores  for AI models implicitly relies on an underdeveloped theory of capability. This often obscures true model performance. The authors advocate for treating AI evaluations as formal inference tasks grounded in explicit capability theories drawn from psychometrics, such as IRT and CDM. To support this position, the paper provides a conceptual framework adapting these human-centric theories to account for systematic variations in AI behavior. The authors also demonstrate how different theoretical assumptions yield divergent capability estimates. The paper concludes with practical guidelines for evaluators to consciously design and report their evaluation assumptions.

**Position:**

Yes

**Position In Title:**

Yes

**Related Work:**

2

**Strengths And Weaknesses:**

Strengths:

1. The paper clearly states its position, arguing that AI evaluations must be grounded in explicit theories of capability. This position is consistently maintained and thoroughly explored.

2. The authors effectively introduce various established psychometric frameworks such as CTT, IRT, CDM, and BNSM into their theories and experiments. And the experiment results provide a foundation for their arguments.

3. As the community faces the limits of standard benchmarks and the unreliability of LLM evaluations, proposing a more solid, theory-based approach is highly relevant and likely to encourage meaningful debate."

Weaknesses:

1. While the paper successfully demonstrates that different capability theories (CTT, IRT, CDM, BNSM) yield different estimates and rankings, it falls short of providing a concrete framework for validating which theory best represents a model's true capabilities. Merely stating that evaluators must choose based on "What do you want a higher score to mean?" is subjective. This may cause evaluators pick the psychometric model that makes their AI system look best.

2. The empirical proof-of-concept focuses solely on input sensitivity (prompt perturbations). However, Equation 6 introduces other critical confounders such as hyperparameters and context. The paper would be significantly stronger if it demonstrated how an explicit theory of capability could show these specific compound uncertainties, rather than limiting the experiments to just prompt phrasing on simple tasks.

3. The Alternative Views section overlooks that the scalability and operational overhead of adopting complex theories like BNSM or CDM. Manually designing item-to-skill mappings or gathering calibrated parameters for IRT across massive, multi-domain benchmarks requires immense effort. The paper does not adequately address how the community can practically scale these theory-driven evaluations for general-purpose LLMs without paralyzing iteration speed.

**Support:**

3

---

> ### Author Rebuttal · Authors · 2026-03-31
>
> ## W1 and Q1
> The reviewer raises an important concern, and we appreciate the opportunity to clarify our claim.  Our paper is not arguing that evaluators should be free to choose any theory post hoc, nor that theory choice is unconstrained. The point is instead that **evaluation already depends on theoretical assumptions, even when those assumptions are hidden** behind simple metrics such as accuracy.
>
> Our proposal is for transparency and discipline: evaluators should state the construct they aim to measure, justify the assumptions that connect benchmark performance to that construct, and report results in a way that makes those commitments visible. Psychometrics is valuable here not because it supplies a single correct model, but because it has long studied how different models encode different assumptions about ability, task structure, and noise.
>
> We agree that selective reporting is a real risk. In a revision, we will make this more explicit and add practical safeguards, such as:
> (1) Pre-specifying the capability construct and modeling assumptions before running the evaluation,
> (2) Distinguishing confirmatory from exploratory analyses, and
> (3) Where feasible, reporting sensitivity across nearby plausible capability models rather than only the most favorable one.
>
> More broadly, our aim is not to license “theory shopping,” but to make theoretical commitments auditable. **Hidden assumptions are easier to manipulate than explicit ones.**
>
>
> ## W2 and Q2
> We agree that prompt perturbations are only one source of uncertainty among several discussed in the paper. Our reason for focusing on them in the proof-of-concept was to isolate a common and well-documented source of systematic variation while keeping the empirical demonstration compact.
>
> **The broader argument does not depend on prompt perturbations specifically**, but we used it to represent a more general point: benchmark performance is not a direct measurement of capability unless the evaluator specifies which sources of variation are being treated as noise, which are being averaged over, and which are being held fixed.
>
> Since this is a position paper, our empirical goal was not to exhaustively quantify every confounder, but **to show with one clear case that different capability assumptions can lead to different inferences** from the same benchmark evidence.
>
>
> ## W3
> We agree that scalability is an important practical constraint, especially for large, multi-domain evaluations. We briefly address this point in the first Alternative View, where we discuss the value of fast/cheap iteration. We also outline the additional effort involved in IRT, CDM, and BNSM models in Section 6, Aspect 3 (data considerations).
>
> **Our position is not that all evaluations should adopt the most elaborate capability model available**. In fact, we do not see CTT-style aggregation and richer models such as IRT, CDM, or BNSM as mutually exclusive. Simpler aggregate evaluations may remain appropriate for rapid iteration or low-stakes monitoring, while more theory-rich evaluations may be especially appropriate for scientific comparison, auditing, or high-stakes claims about progress, safety, or capability. In other words, the practical question is not whether the community should replace all benchmarks with BNSMs, but when a given use case warrants more explicit modeling assumptions. We will sharpen this point in a revised version.
>
> ## Q3
> Thank you for the suggestion! We believe the answer to this question is mainly about taste. The general point we’re trying to make is that **evaluations is an exercise in deciding which assumptions a practitioner finds reasonable about the data generating process**, within the budget constraints they have. This is more of a philosophical exercise than a data-driven one, and as such our “framework” is a set of guiding principles for practitioners, rather than an overly prescriptive procedure of what everyone should do. There is no “ground truth” in what one could do – if one does not believe that BNSM is an appropriate model, then the practical data considerations are moot to begin with. But who is to say that one theory is better than the other? There is certainly no consensus even in human cognitive models/testing.
>
> Say we did suggest some prescriptive guidelines: given some costs c, budget b, at what threshold should an evaluator choose the more comprehensive model? It will likely be some ratio of the two, but quantifying that threshold is purely taste-based and depends on the normative value one assigns to adhering to a theory of capability.
>
> These questions are certainly important, and perhaps future work may develop some consensus on general best practices. But in sum, 1) we feel proper treatment of this question is a separate paper on its own; and 2) we do not believe that being too prescriptive on the normative values people should have is the correct approach, hence why we propose a set of guiding principles instead of rigid decision rules.

---

> > ### Author Rebuttal · Reviewer_xFze · 2026-04-03
> >
> > Thank you for the thoughtful rebuttal. I appreciate the clarifications, especially the emphasis on making theoretical commitments explicit and auditable rather than encouraging post hoc ''theory shopping''. That said, my main concern remains: the paper still does not offer a sufficiently concrete way to constrain or validate theory choice in practice, and the response to the scalability and decision-framework questions remains intentionally high level. I understand that the authors view these issues as partly normative, but for me this also limits the paper’s practical contribution beyond the core conceptual argument. Overall, I still find the paper interesting and potentially valuable for discussion, but I do not feel the rebuttal fully resolves the key concerns I raised, so I am maintaining my original score

---

### Decision · Program_Chairs · 2026-04-30

**Decision:**

Accept (regular)

**Comment:**

Reviewers generally agreed that the topic is important to the community and that the paper offers a useful conceptual reframing, supported by both qualitative discussion and a proof-of-concept empirical analysis showing that different capability theories can yield different interpretations of the same benchmark results.

Even the more critical review acknowledged the importance of the core issue, with its concerns focused less on whether the position matters and more on scope, practical scalability, and how prescriptive the paper should be in guiding evaluator choices. I found the rebuttal helpful on these points: it clarified that the paper is not advocating unrestricted “theory shopping,” but rather asking evaluators to make already-implicit assumptions explicit and auditable. While the empirical demonstration is necessarily limited and the practical guidance remains more principled than fully operational, I view that as acceptable for a position paper whose main contribution is to reframe how the community should think about benchmark-based evaluation. Overall, I recommend acceptance.